# Past local summer temperature changes over the past 440 ka revealed by Total Air Content in the Antarctic EPICA Dome C ice core

Dominique Raynaud[1*], Qiuzhen Yin[2*], Emilie Capron[1*], Zhipeng Wu[2], Frédéric Parrenin[1], André Berger[2], Vladimir Lipenkov[3]

[1]Institute of Environmental Geoscience (IGE), Université Grenoble Alpes, CNRS, IRD, INRAE, Grenoble-INP, 38000 Grenoble, France
[2]Earth and Life Institute, Earth and Climate Research Center, Université catholique de Louvain, 1348 Louvain-
la-Neuve, Belgium
[3]Climate and Environmental Research Laboratory, Arctic and Antarctic Research Institute, Saint Petersburg, 199397, Russia

*The authors contributed equally to this study.

*Correspondence to*: Dominique Raynaud (dominique.raynaud2@univ-grenoble-alpes.fr) & Emilie Capron (emilie.capron@univ-grenoble-alpes.fr)

**Abstract.** Seasonal temperature reconstructions from ice cores are missing over glacial-interglacial timescales, preventing a good understanding of the driving factors of Antarctic past climate changes. Here the total air content (TAC) record from the Antarctic EPICA Dome C ice core is analyzed over the last 440 thousand of years (ka).
While the water isotopic record, a tracer for annual mean surface temperature, exhibits a dominant ~100 kyr cyclicity, the TAC record is associated with a dominant ~40 kyr cyclicity. Our results show that the TAC record is anti-correlated with the mean insolation over the local astronomical summer half-year. They also show for the first time that it is highly anti-correlated with local summer temperature simulated with an Earth system model of intermediate complexity. We propose that (1) the local summer insolation controls the local summer temperature,
(2) the latter, through the development of temperature gradients at the near surface of the ice sheet (<2m), is affecting the surface snow structure and (3) those snow structure changes propagating down to the bottom of the firn through densification are controlling eventually the pore volume at the bubble close-off and consequently, the TAC. Hence, our results suggest that the EDC TAC record could be used as a proxy for local summer temperature changes. Also, our new simulations show that the mean insolation over the local astronomical summer
half-year is the primary driver of Antarctic summer surface temperature variations while changes in atmospheric

greenhouse gas concentrations and northern hemisphere ice sheet configurations play a more important role on Antarctic annual surface temperature changes.

## 1 Introduction

The analysis of Antarctic ice cores provides paramount information to reconstruct and understand the climate dynamics of the past 800 thousands of years (hereafter ka). Amongst the key climatic parameters that can be inferred from these deep ice cores are the local mean annual temperature reconstructed from the isotopic composition of ice, e.g. $\delta D$ (Dome Fuji Ice Core Project Members, 2017; Jouzel et al., 2007), and past atmospheric greenhouse gas (GHG) concentrations measured on air trapped in the air bubbles (Bereiter et al., 2015; Loulergue et al., 2008; Lüthi et al., 2008). However, the climate during local summer, which is a critical season for polar regions especially in terms of solar energy received, is seldomly discussed, except through the highlighting of local insolation signatures on the $O_2/N_2$ ratio of trapped gas (Bender, 2002; Kawamura et al., 2007; Landais et al., 2012) and the air content in bubbles (Eicher et al., 2016; Epifanio et al., 2023; Lipenkov et al., 2011; Raynaud et al. 2007) in ice core records. In particular, there is no suitable proxy of the local summer temperature. Moreover, a debate remaining in Antarctic climate study is related to whether the Antarctic temperature variations on orbital timescales are controlled by the Northern Hemisphere (NH) insolation or by local insolation (Huybers and Denton, 2008; Kawamura et al., 2007). As the insolation over Antarctica is received mostly during summer, having a proxy of summer temperature would therefore be essential for helping to decipher the role of NH versus local insolation as well as the role of other glacial boundary conditions such as the changes in atmospheric GHG concentrations, in Southern Ocean sea ice extent and in ice sheet configuration.

An imprint of local insolation changes has been evidenced in tracers which are measured in the air trapped in polar ice core. Indeed, air bubbles close-off from the surrounding atmosphere and they become trapped in ice, an air tight material resulting from the densification and diagenesis of the snow deposited at the surface. These processes of densification and diagenesis take place in the upper layer at the surface of the ice sheet (between 60-120 m-deep typically), which is characterized by an open porosity to the atmosphere and by two successive stages, snow and then firn, associated with different densification regimes (Anderson and Benson, 1963). In the absence of surface melting, which is the case at the EPICA Dome C (EDC) site (75°06' S, 123°21' E; 3233 m a.s.l.) on the high plateau of East Antarctica, and according to the ideal gas law, the amount of air (V) in the bubbles at close-off depends on their physical volume ($V_c$) and on the pressure ($P_c$) and temperature ($T_c$) of the air contained in $V_C$ at the enclosure time (Martinerie et al., 1992). In first approximation, $T_c$ is equivalent to

the mean annual temperature prevailing at the surface of the ice sheet, which is estimated from the isotopic composition of the ice.

Then, V can be defined as the total volume of air in unit mass of ice, measured at standard temperature $T_0$ and pressure $P_0$, and $V_c$ is the pore volume per unit mass at close-off:

Equation (A)           $V = V_C \, P_c/T_c \times T_0/P_0$

Furthermore, the porosity at close-off, Vc, is related to temperature.

During previous works V (for air content) and TAC (for Total Air Content) have been interchangeably used for designating the same property. In this work we are using TAC, following other recent studies.

Long-term high resolution studies of TAC records obtained from several deep ice cores in central Antarctica (Kawamura et al., 2001; Lipenkov et al., 2011; Martinerie et al., 1994; Raynaud et al., 2007) revealed

a long-term large variability that cannot be explained by changes in $T_c$ or $P_c$. In particular at EDC, about 85% of the variance observed in the high-resolution V record covering the last 440 ka can be explained neither by $P_c$, nor by $T_c$ changes. This led to consider that other properties, beside the mean annual temperature and barometric pressure at the surface, may also influence TAC. By using continuous wavelet transform (CWT) analysis, Raynaud et al. (2007) found that the EDC TAC record shows significant power in the obliquity and precession

bands, with a dominant ~40 ka signal, which has been assumed to reflect orbitally-driven changes in local summer insolation.

To account for the observed anti-correlation between local summer insolation and TAC, a mechanism has been proposed (Lipenkov et al., 2011; Raynaud et al., 2007) where the local summer insolation, by controlling the near-surface (<2m) snow temperature and temperature gradients during summertime, affects the near-surface

snow structure and consequently the porosity of the firn pores at close-off, i.e. the TAC of air bubbles  (Lipenkov et al., 2011; Raynaud et al., 2007). Based on such assumption, TAC was used as an orbital dating tool to constrain ice core chronologies. TAC-based age markers have been used for the latest official chronologies for polar ice cores, AICC2012 (Bazin et al., 2013; Veres et al., 2013) and AICC2023 (Bouchet et al., 2023). However, the exact physical processes that lead to an imprint of the local summer insolation in the TAC record are unclear. In

particular, uncertainties remain regarding the link between TAC and the surface climatic parameters such as local temperature.

In this study, we use the TAC record measured in the EDC ice core covering the last 440 ka (Raynaud et al., 2007). We compare it with a new local insolation index and with transient simulations performed with the Model LOVECLIM1.3 to explore the link between insolation, Antarctic summer temperature and TAC, as well

as the related mechanisms. Finally, we compare the TAC record with the EDC δD record to understand the major driving factors of the summer and annual mean temperature changes in Antarctica.

## 2 Method

### 2.1. TAC measurements

TAC measurements have been performed at the Institute of Environmental Geosciences (Grenoble, France) using an original barometrical method implemented with an experimental setup called STAN (Lipenkov et al., 1995). Results of the numerical data of the measurements versus depth along the EDC ice core can be found in the appendix in Raynaud et al. (2007). The TAC data shown on Fig. 1 are displayed on the AICC2023 ice age chronology (Bouchet et al., 2023).

### 2.2. Model and simulations

The model used in this study is LOVECLIM1.3, a three-dimension Earth system Model of Intermediate Complexity (EMIC), with its atmosphere (ECBilt), ocean and sea ice (CLIO) and terrestrial biosphere (VECODE) components being interactively coupled (Goosse et al., 2010). The model setup is the same as the one used in Yin et al. (2021) and a detailed description can be found there. In terms of the Antarctic climate, LOVECLIM1.3 reproduces reasonably the spatial pattern and the magnitude of surface temperature over Antarctica in winter, summer and annual mean (Fig. 2a). It is slightly cooler in western Antarctica in the model probably related to its rough resolution. The seasonal temperature cycle at the EDC site (Fig. 2b) as well as the Antarctic inversion (Fig. 2c) are also well reproduced by the model.

Although LOVECLIM1.3 is classified as an EMIC model, its complexity is high for this kind of models and its ocean component is a full general circulation model, so it remains challenging to run full transient simulations with this model. We therefore first performed a transient simulation with 10x acceleration covering the last 800 ka, which allows to compare the simulated local summer temperature with the TAC record over the entire last 440 ka. In this simulation, the variations of orbital forcing and GHG were considered, and the global ice sheets were fixed to their pre-industrial condition. Using the same model and the same acceleration technique, it has been shown that 10x acceleration has a significant impact on deep ocean temperature, but it has no major impact on surface temperature (Yin and Berger, 2015). This is further confirmed in our study where the Antarctic summer and annual mean temperature changes of the 10x acceleration simulation are matching well with that of the non-accelerated simulations (Supplementary Fig. S1).


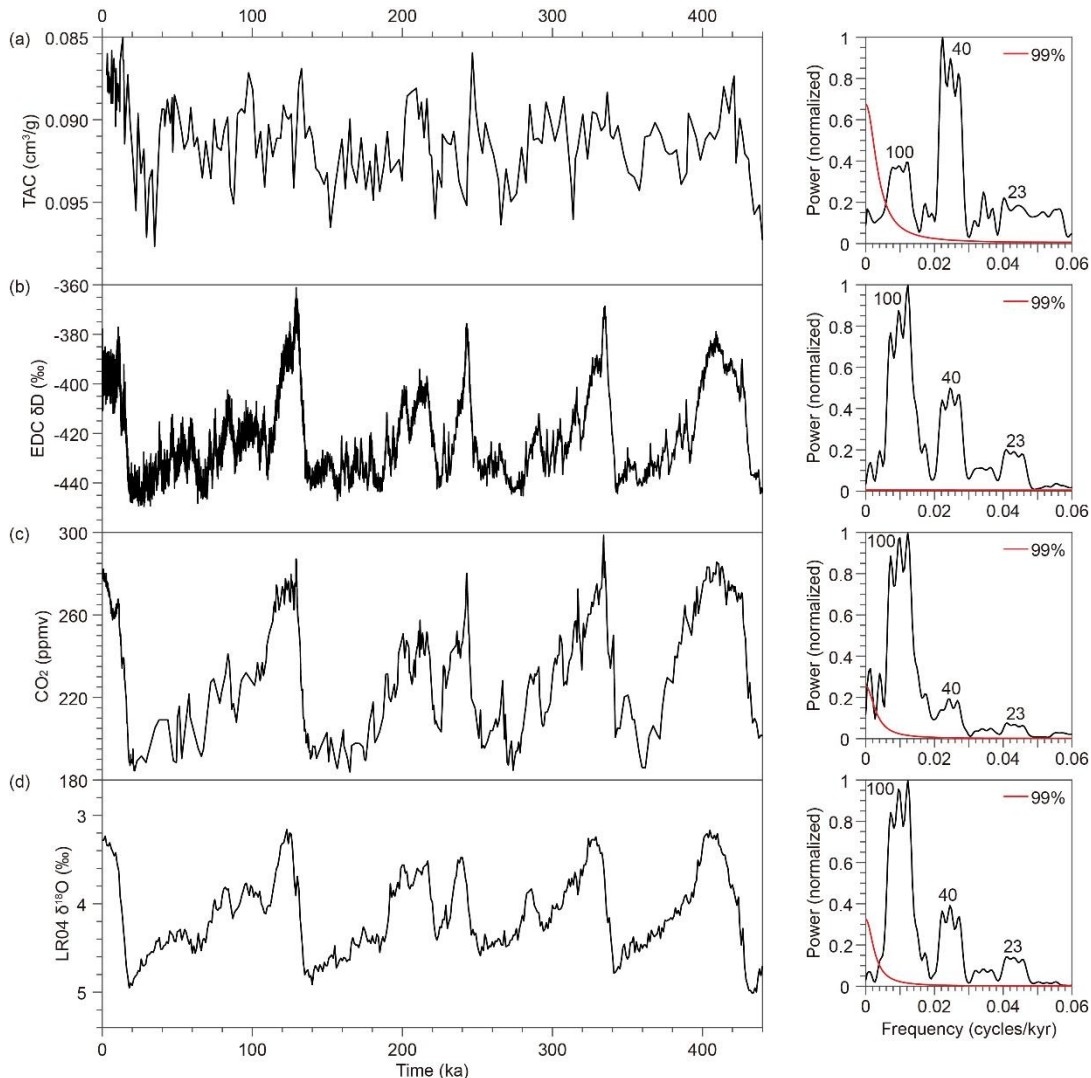

**Figure 1. Variations and spectra over the past 440 ka of (a) TAC record (raw data, Raynaud et al. 2007), (b) EDC δD record (Jouzel et al., 2007), (c) CO₂ concentration (Lüthi et al., 2008) and (d) benthic δ¹⁸O (Lisiecki and Raymo, 2005). The major periodicities in kyr are indicated. The EDC TAC record, δD record and CO₂ concentrations are displayed on the AICC2023 chronology (Bouchet et al., 2023). The spectra are calculated using the multi-taper method (MTM), the number of tapers is set to 2 and the zero padding is set to 5, and the 99% confidence limit is shown (red curves).**


We further performed transient simulations without acceleration for some glacial-interglacial episodes of the last 440 ka to investigate the relative effects of insolation, GHG and NH ice sheets (see section 5 for

simulation periods and results). Each episode includes three simulations. The first two simulations, Orb and OrbGHG, were performed in Yin et al. (2021) and a detailed description of the experiment setup can be found there. Here we only give some brief introduction. In the Orb simulation, only the change of orbital forcing (Berger and Loutre, 1991) was considered, with the GHG and ice sheets being fixed to their pre-industrial condition. In the OrbGHG simulation, the change of GHG (Loulergue et al., 2008; Lüthi et al., 2008; Schilt et al., 2010) was

considered in addition to the orbital forcing. In the third simulation, OrbGHGIce, the change of NH ice sheets (Ganopolski and Calov, 2011) was additionally considered, but the Southern Hemisphere ice sheets remained fixed to the pre-industrial conditions. The initial conditions were provided by a 2000-year equilibrium experiment with the NH ice sheets, GHG concentrations and astronomical parameters at the starting date of the simulated period. In the presence of land ice, albedo, topography, vegetation and surface soil types corresponding to ice-

covered conditions were prescribed at corresponding model grids in LOVECLIM1.3. Detailed description of the ice sheet setup can be found in Wu et al. (2023).

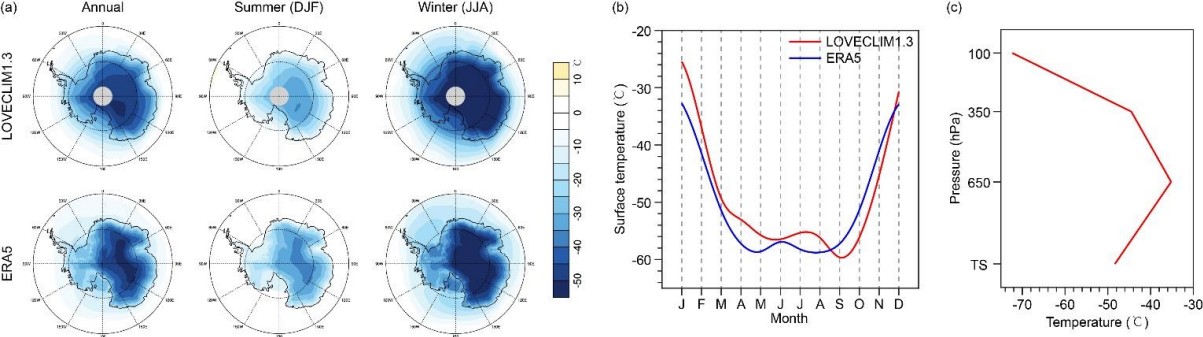

**Figure 2. Comparison of the 1971-2000 mean climate in Antarctica simulated by LOVECLIM1.3 and the ERA5 reanalysis (https://cds.climate.copernicus.eu/cdsapp#!/home) for mean annual, summer (DJF) and winter (JJA) surface**
**temperature (a) and for the seasonal cycle of the surface temperature at the EDC site (b), as well as the simulated vertical mean annual surface temperature profile at the EDC site.**

## 3 EDC TAC changes vs. Local summer insolation variations over the past 440 ka

The spectral analysis of the TAC record shows that its variations over the last 440 ka are dominated by the 40-kyr cycle (Fig. 1), which corresponds to the main periodicity of obliquity. It also shows the 100-kyr cycle as well as

the 23-kyr and 19-kyr cycles which correspond to the precession cycles (Berger, 1978), but their amplitude in the

power spectrum is much weaker. Overall, this spectral characteristic illustrates that the variations of TAC are strongly correlated with the astronomical forcing and could be linked to insolation changes (Lipenkov et al., 2011; Raynaud et al. 2007).

When comparing a proxy record with insolation, it is not necessarily straightforward to decide which insolation index to choose, because different insolation metrics exist and their relationship with climate is not always clear (Berger et al., 2010; Berger et al., 1993). In a previous study presenting the EDC TAC record over the past 440 ka (Raynaud et al., 2007), an Integrated Summer Insolation index (also referred as ISI) was established and compared with the TAC record in order to find the ISI curve with variations that would resemble the most the changes recorded in the TAC record. The ultimate objective of such exercise was to identify an insolation target to infer dating constraints from TAC based on orbital tuning. The most appropriate orbital tuning target was found by tuning the precession-to-obliquity amplitude ratio of the insolation index on the corresponding spectral signature of the TAC record. It corresponds to the so-called ISI 380 curve that was obtained by summing over the year, the daily insolation above a threshold of 380 W.m$^{-2}$. Hence, this orbital tuning heavily relies on (1) the tuning of the relative amplitudes of the precession and obliquity in the power spectra and as a consequence (2) the selected insolation threshold value. It is also based on the assumption of a time-linear (constant) response of TAC to the selected insolation threshold. To avoid these assumptions, we propose to use a simpler and independent insolation index in the present work: the mean insolation over the astronomical half-year summer at 75°S (the latitude of EDC).

The astronomical summer half-year in the Southern Hemisphere (SH), which corresponds to the astronomical winter half-year in the NH, is defined as the time interval during which the Earth travels from fall (September) equinox to spring (March) equinox on the ecliptic (Berger and Loutre, 1994; Berger and Yin, 2012; Berger et al., 2024). The advantage of using astronomical season is to allow for the change of the length of seasons. The astronomical summer half-year in the SH is the main interval during which the southern polar regions (regions within the Antarctic Circle) receive solar radiation over a year. The length of this half-year summer is varying in time and is only a function of precession (Berger and Loutre, 1994; Berger and Yin, 2012; Berger et al., 2024). Over the last 440 ka, it varies between 171.3 and 194.0 days. The total solar radiation received over the half-year astronomical summer is only a function of obliquity (Berger et al., 2010). Therefore, the mean summer insolation at 75°S, which is calculated by dividing the total irradiation received during the half-year summer by its length, is both a function of obliquity and precession with obliquity being dominant. Compared to the total summer insolation or the integrated insolation above a threshold (ISI), the mean insolation of the astronomical summer half-year (it is referred as mean summer insolation hereafter) considers not only the total amount of energy

received during the astronomical summer half-year, but also the length of the astronomical summer half-year which could also be important.

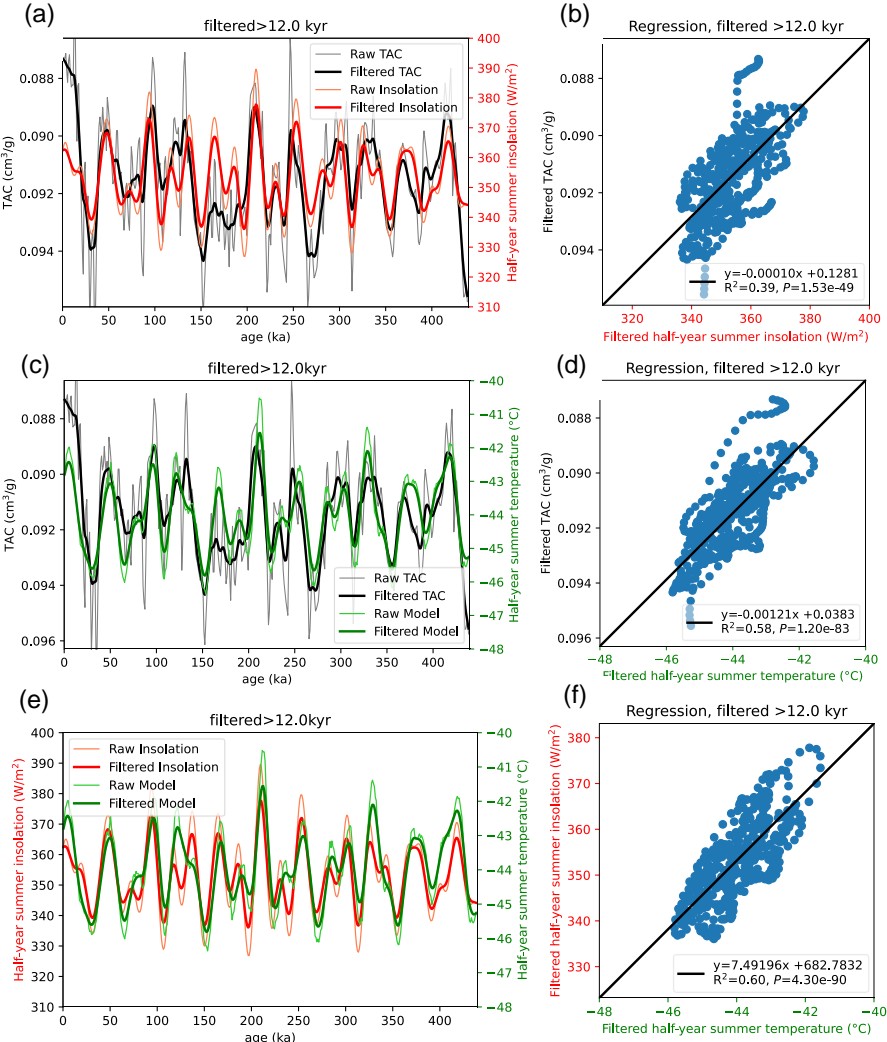

Figure 3. Comparison of TAC record with (a) mean insolation during astronomical half-year summer at 75°S calculated using the solution of Berger and Loutre (1991) and with (c) simulated mean half-year summer (October to March) temperature at EDC. Their corresponding linear regression analyses are shown in (b) and (d). (e) Comparison of half-year summer insolation with simulated mean half-year summer temperature and corresponding linear regression in (f). Low-pass filtered >12 kyr is applied on the TAC, insolation and summer temperature raw data before comparison. Note that the y axis for TAC is reversed on the left panels to ease the visual comparison.

Fig. 3 shows the comparison between TAC and the mean summer insolation. Since we focus here on the orbital-scale variations, a low-pass filter (> 12 kyr) has been applied to the TAC data before the comparison in order to eliminate the high-frequency signals. A good resemblance in term of temporal structure and amplitude is observed between the two variables and the two datasets appear well anti-correlated with a $R^2$ correlation coefficient of 0.39. This comparison is surprisingly good considering that TAC could also be influenced by other factors.

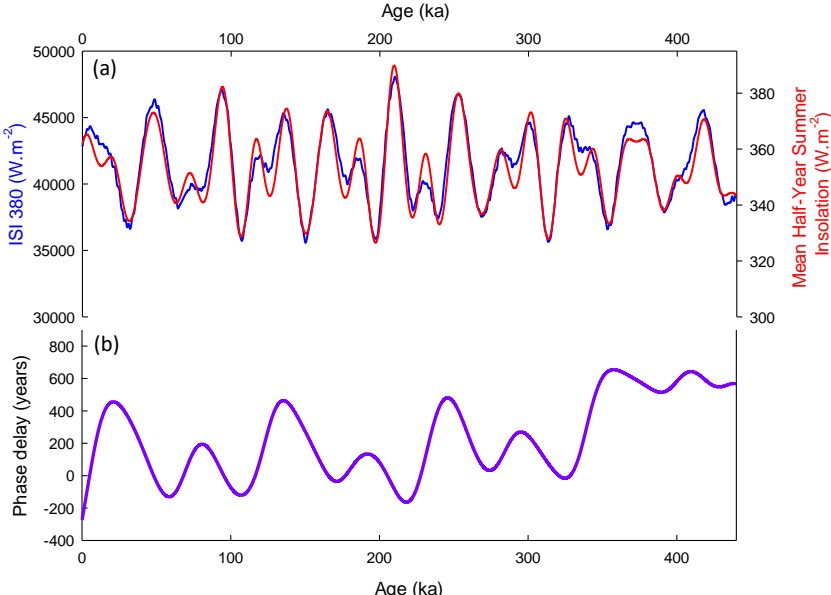

**Figure 4. (a) Comparison of ISI 380 (blue) with the mean insolation during astronomical half-year summer (red), (b) evolution of the phase delay (purple) between the two insolation curves filtered in the 15-46 kyr band. ISI 380 is represented here as a flux following the definition given in Eicher et al. (2016).**

This new result implies that for dating purposes the mean summer local insolation is more appropriate than the ISI 380 curve (Raynaud et al., 2007) to be used as an orbital dating target. Indeed, it appears preferable to favor the mean summer insolation record as it is fully independent from the TAC record compared to ISI 380 although the degree of anti-correlation with the TAC record is of similar magnitude. While this is beyond the scope of this study to discuss in details the implications for the definition of TAC-based age markers to constraint the EDC ice core dating (Bouchet et al., 2023), we present in Fig. 4, a comparison of the two insolation indexes over the past 440 ka. We observe a strong resemblance in terms of the relative amplitude and timing of changes in the two records. Following the approach described in Raynaud et al. (2007), we calculate the evolution of the

phase delay between the two records filtered in the 15-46 kyr band to provide a quantification of the age differences that could be generated from the use of one curve or the other for orbital dating purposes. On average, the age difference is of about 260 years and never above 650 years. These age differences should be considered as minimal as they do not account for other sources of age uncertainties when building a TAC-based orbitally-tuned chronology e.g. our ability to define precisely the tie points between the TAC data and the insolation index also depends on the quality of the visual resemblance between TAC and the insolation target. These matters will be fully discussed in a subsequent study.

**4 EDC TAC changes vs. local summer temperature changes simulated by the LOVECLIM1.3 model**

While it appears that the EDC TAC record is anti-correlated with the local mean summer insolation, its relationship to seasonal surface temperature reconstructions has not been investigated yet. Recently, a quantitative reconstruction of the seasonal temperature changes in West Antarctica has been produced throughout the Holocene (Jones et al., 2023), but to our knowledge no seasonal temperature reconstructions in Antarctic ice cores are available over the longer glacial-interglacial timescale. Hence, we propose to compare the EDC TAC record with the local summer temperature changes obtained from transient simulations performed with LOVECLIM1.3. The comparison of the LOVECLIM1.3 simulated summer temperature in West Antarctica with the one reconstructed by Jones et al (2023) shows that they compare well in both the trend and the magnitude of temperature change over the Holocene, both showing that the summer temperature in west Antarctica had an increasing trend from early to mid-Holocene and reached a maximum at ~4 ka BP followed by a decreasing trend (Supplementary Fig. S2). This validates the LOVECLIM1.3 simulations in reproducing past summer temperature changes in West Antarctica.

Fig. 3 shows that the EDC TAC values increase when the modeled local summer temperatures decrease. It also shows that there is a high and positive correlation between the simulated summer temperature and the mean summer insolation. Linear regression analysis shows that TAC and the simulated summer temperature are highly and negatively correlated. The linear coefficient of determination ($R^2$=0.58 when a low-pass filter > 12 kyr is applied on the TAC data) indicates that about 58% of the TAC variability observed on the EDC ice core over the last 440 ka is explained by the half-year summer temperature at EDC, which suggests that summer temperature variations in the central part of East Antarctica can be considered as the main driving forcing factor of TAC. The slope of the regression shows an increase of 0.0012 cm$^3$ of TAC per gram of ice for a cooling of 1°C of the mean half-year summer local temperature at EDC. The regression analysis has been also evaluated by using the raw

TAC data and a low-pass filter of > 6 kyr. The good anti-correlation between TAC and the summer temperature is not altered although the regression slope is slightly affected with an increase of about 0.0011 cm$^3$ of TAC per gram of ice for a cooling of 1°C. Hence, we propose that the link between summer insolation and TAC variations exists through the summer temperature changes (see also section 6). Indeed, as proposed in previous studies (Lipenkov et al., 2011; Raynaud et al., 2007), the local summer insolation controls the near-surface snow temperature and the vertical temperature gradients in snow. In turn, the latter could affect the near-surface snow structure and consequently the porosity of the firn pores at close-off, i.e. the TAC. The good correlation between the two independent climate variables, TAC measured from ice cores and summer temperature simulated by the LOVECLIM1.3 model, indicates that the EDC TAC record can be used as a proxy for local summer temperature. The relationship between the TAC record from the EDC ice core and local summer temperature should be further investigated in other ice cores from Antarctica and Greenland.

In our study, we did not account for variations in surface elevation. In principle, during glacial periods, the reduced surface accumulation rate leads to lower surface elevation (Raynaud et al., 2007). But there are also dynamical effects which make these reconstructions of surface elevation uncertain. Surface elevation changes affect our study in two different ways. First, TAC should be corrected for atmospheric pressure changes to get a record of porosity at close-off. Some of these atmospheric pressure changes are due to variations in surface elevation, another part might be due to change of atmospheric conditions (like the temperature of the air column). Because the reconstruction of surface elevation changes are uncertain, we have chosen to not correct for this effect. Second, surface elevation changes should also be ideally taken into account in our LOVECLIM1.3 simulations of summer or annual temperature changes at EDC. Unfortunately, in our climate model, the Antarctic temperature is fixed and we did not account for surface elevation changes. We could apply an a posteriori correction for surface elevation changes, but because the temperature variations (either annual or in summer) are probably under-estimated, applying this a posteriori correction would have a too strong influence. Finally, we should note that these two elevation corrections (for TAC and for the climate model) go in the same direction: during glacials, a corrected summer temperature would be warmer, and the corrected TAC of the ice would be smaller, so there is a chance that these two corrections would cancel each other and that the overall correlation between TAC and modeled summer temperature would not be much affected.

A wavelet analysis (Fig. 5a-b) shows that the variations of both TAC and the simulated summer temperature are dominated by the ~40-kyr cycle throughout the last 440 ka, indicating the major role of obliquity. They also contain a ~20-kyr cycle, but this cycle is not stable in time, with an amplitude which is relatively strong for instance around 100 and 200 ka in both TAC and simulated summer temperature but weak during other periods.

This may be related to the amplitude modulation of eccentricity on precession (Berger and Loutre, 1991). The eccentricity at ~100, 200 and 300 ka was large, leading to large variations of precession and thus stronger effect of precession around these time intervals than during other times. However, the power of obliquity is generally more important than the one of precession when considering the past 440 ka, in particular around 400 ka and during the last 50 ka when eccentricity was small leading to small variations of precession. The 100-kyr periodicity is also observed in both TAC and the simulated summer temperature but with a weak amplitude. This weak signature of the 100-kyr cycle in both the TAC and the mean half-year summer temperature must not arise from the mean summer insolation because there is no 100-kyr cycle in the mean insolation (Fig. 5c). It must arise from the glacial-interglacial boundary condition changes which are characterized by a major periodicity of 100 kyr (see section 5).

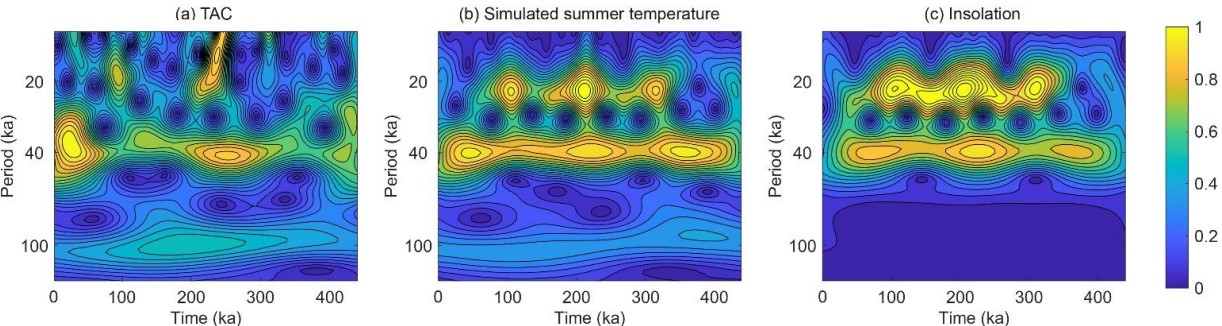

**Figure 5. Continuous wavelet transforms of (a) the low-pass (> 12 kyr) filtered TAC record, (b) simulated mean half-year summer temperature at the EDC site and (c) mean insolation during astronomical summer half-year at 75°S.**

## 5 Deciphering the driving forcing of past Antarctic summer and annual temperature changes

The δD record from Antarctic ice has been widely used as an important proxy for mean annual precipitation-weighted condensation temperature over Antarctica (Jouzel et al., 2007; Stenni et al., 2010), although it has also been suggested to be biased toward winter temperatures (Laepple et al., 2011). We use it in the present work as a record of the EDC mean annual temperature. Different from the TAC record which is dominated by the 40-kyr cycle, the EDC δD record is dominated by the ~100-kyr cycle (Fig. 1). The difference in the dominant periodicities between the TAC and δD records suggests that the major driving factors for the summer and annual mean temperatures are different. As shown above, TAC is strongly linked to local summer temperature which is mainly controlled by the local summer insolation. However, the dominant 100-kyr cycle in

the δD record suggest that the annual mean temperature is mainly controlled by the glacial-interglacial boundary conditions such as global ice volume and GHG which are dominated by strong 100-kyr cycle (Fig. 1).

    To investigate the response of Antarctic climate to insolation, GHG and ice sheets, the three sets of transient simulations, Orb, OrbGHG, and OrbGHGIce (see section 2) which cover the last five interglacial-glacial episodes, are analyzed here. We first compare the δD-based temperature reconstruction with the simulated annual

mean temperature of OrbGHGIce (Fig. 6). We observe that this comparison is quite good over the simulation periods in terms of climate variation pattern, showing the capacity of the model to simulate the orbital-scale climate variations at the EDC site. One may also note that the magnitude of the temperature change between glacial and interglacial is significantly underestimated in the model as compared to the reconstruction. In our simulation, the Antarctic ice sheet as well as the sea level are kept invariant, which could contribute at least partly

to the underestimated amplitude of temperature change in the model. However, a recent study using borehole thermometry and firn properties suggests that the temperature reconstruction using water-stable isotopes calibrated against modern spatial gradients could generate a too large amplitude of glacial-interglacial temperature change (Buizert et al., 2021). For example, at EDC, the Last Glacial Maximum (~26-18 ka) temperature relative to the pre-industrial time is about -9°C according to the δD-based reconstruction (Jouzel et al., 2007), but it is

only -4.3±1.5°C in a more recent reconstruction (Buizert et al., 2021). In our simulation, the EDC annual temperature at 17 ka is -1.7°C relative to the pre-industrial era, and the simulated largest glacial-interglacial amplitude is ~3°C, which is within the uncertainty of the recent reconstruction (Buizert et al., 2021). Nevertheless, such small glacial-interglacial difference found in our study seems difficult to be explained taking into account the relationship between snow accumulation rate and surface temperature from the saturation vapor relationship

(Cauquoin et al., 2015). However, what is essential for our study is that the model could capture the orbital-scale variability of the reconstructed temperature.

    To investigate the relative effect of insolation, GHG and NH ice sheets on the summer and annual temperature at EDC, we take the 133-75 ka period, which includes the last interglacial period and the glacial inception, as an example to compare the Orb, OrbGHG and OrbGHGIce experiments (Fig. 7). As expected, the

annual and summer temperatures at EDC are reduced in response to reduced GHG and increased NH ice sheets (Fig. 7b, 7d), with the impact of GHG being larger than the NH ice sheets for both simulated temperatures. As far as the summer temperature is concerned, the variation pattern of OrbGHG and OrbGHGICE is very similar to the one of Orb (Fig. 7b), showing the minor effect of GHG and NH ice sheets on the temporal variations of summer temperature. Over this period, the largest summer temperature change caused by insolation is 4.4°C while it is

1°C for GHG and 0.3°C for ice sheets. However, when the simulated annual temperature is considered (Fig. 7d),

its variation pattern is largely altered in response to changes of GHG and NH ice sheets. Over this period, the largest annual temperature change caused by insolation is 1.2°C while it is 1°C for GHG and 0.5°C for ice sheets. These results clearly show that as compared to insolation, GHG and NH ice sheets have relatively weaker effect on the summer temperature but they have relatively stronger effect on annual mean temperature. This, at least

partly, explains why the TAC and δD records display different dominant periodicities over long timescales.

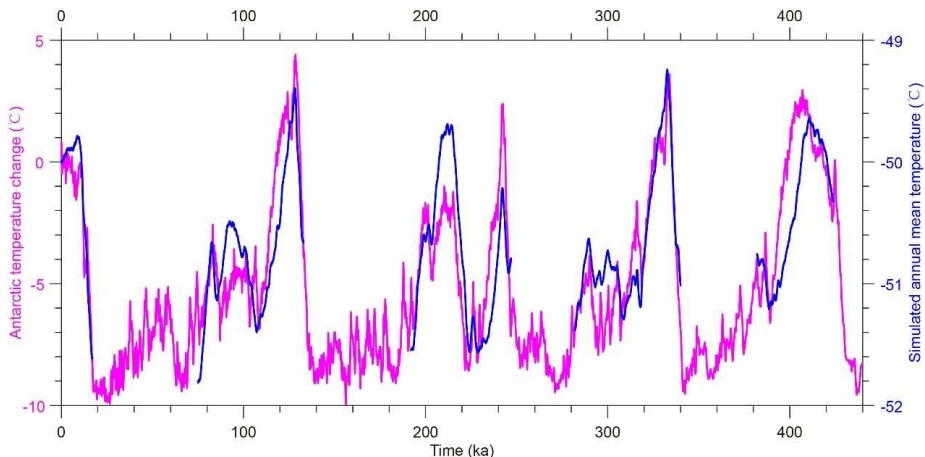

**Figure 6. Comparison of the EDC mean annual temperature record (pink line, Jouzel et al. 2007) with the simulated annual mean temperature of the OrbGHGIce experiments (blue line, this study).**


To better understand the difference between the summer and annual mean temperature, the simulated winter temperature is also analyzed. The wavelet analysis of the half-year winter temperature shows very strong ~20-kyr cycle and an obvious ~100-kyr cycle, but the ~40-kyr cycle is very weak (Fig. 8a). The 100-kyr cycle results most probably from the effect of GHG. The very strong 20-kyr cycle but very weak 40-kyr cycle is quite

intriguing. As far as insolation is concerned, the low-latitude insolation is dominated by the ~20-kyr precession cycle. As the solar energy received in Antarctica is very weak during local winter, the strong 20-kyr cycle in the simulated winter temperature could reflect a strong effect of the low latitude climate on the Antarctica temperature during austral winter, possibly via meridional oceanic and atmospheric heat transport. Fig. 8b shows a high, negative correlation between the simulated EDC winter temperature and precession. This indicates that the EDC

winter temperature is strongly affected by boreal summer insolation in low latitudes (small precession parameter leads to high boreal summer insolation and vice versa). It is also shown in Yin and Berger (2012) that during some interglacials such as Marine Isotope Stages 5e, 15 and 17 which are characterized by strong boreal summer insolation, a strong warming could be induced over Antarctica during austral winter, a warming which is even

stronger than in many other regions due to polar amplification. Similar to what happens to the summer
temperature, orbital forcing plays a dominant role also on the winter temperature at EDC (Fig. 7c). As explained
above, the winter temperature at EDC is actually strongly driven by precession and boreal summer insolation, so
on precession timescale, the orbitally-induced temperature variation in winter is in anti-phase with the summer
temperature which is strongly driven by austral summer insolation. This anti-phase relationship leads to a strong
weakening of the orbital signal especially the precession signal in the mean annual temperature (Fig. 7d), making
the effect of GHG and ice sheets more pronounced and thus leading to strong glacial cycles in the mean annual
temperature.

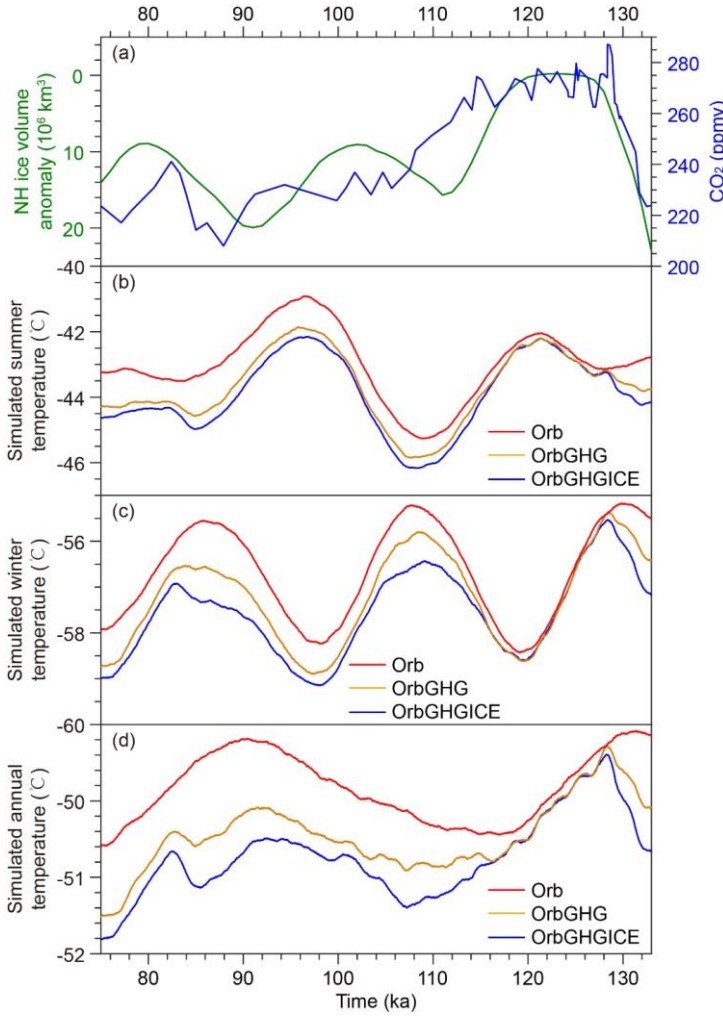

**Figure 7. Effect of insolation, GHG and NH ice sheets on the summer, winter and annual temperature at EDC. (a) CO₂ concentration (blue, Lüthi et al., 2008) and NH ice volume anomaly as compared to pre-industry (green, Ganopolski and Calov, 2011), (b) simulated mean half-year summer (October to March) temperature, (c) simulated mean half-year winter (April-September) temperature, and (d) simulated annual mean temperature from the Orb, OrbGHG and OrbGHGICE experiments. The results of the LOVECLIM1.3 transient simulation without acceleration for the period 133-75 ka are used.**

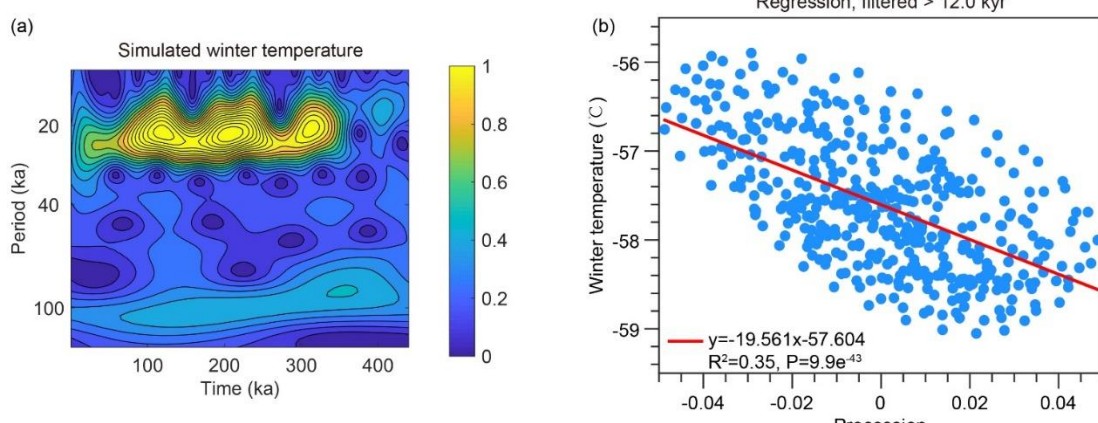

**Figure 8. (a) Continuous wavelet transform of the simulated mean half-year winter temperature at the EDC site from the 10x accelerated OrbGHG simulation, and (b) Correlation between this winter temperature and precession. Low-pass filtered >12 kyr is applied on the winter temperature raw data.**

## 6 Possible mechanisms linking TAC and local summer temperature

The possible mechanism by which summer temperature and near-surface temperature gradients can affect the pore volume at close-off, $V_c$, has been proposed assuming a homogenous firn column and neglecting the sealing effect on the total amount of air trapped in ice (Lipenkov et al., 2011; Raynaud et al., 2007). This simplification seems to be reasonable for low-accumulation sites such as EDC, Vostok, Dome Fuji, because at those sites: (1) the horizontal extent of snow layers characterized by different physical properties, as a rule, does not exceed a few meters, which suggests a patchy pattern of their spatial distribution on and below the ice sheet surface (Ekaykin et al., 2023; Fujita et al., 2009), (2) the variability of density (Hörhold et al., 2011) and microstructural properties (Gregory et al., 2014) of firn is relatively low, as is the stratigraphic-scale variability of the air content of ice (Lipenkov et al., 2011; Lipenkov et al., 1997). In addition, the firn can be affected by layering. Pore closure in denser layers occurs at a shallower depth compared to the pore closure for layers that are less dense. However, in sites with low-accumulation it was shown that regardless of their density (denser or less dense), $V_c$ and hence, $V$ is similar in both types of layers (Fourteau et al., 2019).

The snow metamorphism on the cold Antarctic Plateau is essentially a summertime phenomenon. It speeds up when the temperature of the uppermost layers of snow rises well above the mean annual temperature, thus increasing both the equilibrium concentration of water vapor in the snow pores and the temperature gradients in the near-surface snow. Elevated temperatures and strong temperature gradients promote the rapid growth of snow grains and the formation of a coarse-grained snow structure. Small-scale stratigraphic variations in the snow structure, which are typical in the upper few meters of the snow column, progressively disappear with depth (Alley, 1980), while the average grain size remains related to temperature conditions prevailing at the time of snow diagenesis near the ice sheet surface.

According to the model proposed by Arnaud (1997) for the Antarctic ice sheet, the pore volume at close-off, Vc, should increase with the mean annual surface temperature through the competing densification mechanisms: higher temperature leads to an increase in the relative critical density at the transition between snow and firn (at EDC the critical density is reached at a depth of about 25 m below surface), which in turn implies a greater proportion of the ice-grain edges occupied by pores at close-off, and hence a larger Vc. Our work (Fig. 1) shows a poor correlation and large spectral differences between TAC and mean annual surface temperatures. In contrast we observe a strong anti-correlation between TAC and the simulated mean surface summer temperature. This observation suggests that summer temperature has an inverse effect on Vc compared to the mean annual temperature. Indeed, TAC increases with the ratio of the number of pores to the number of ice crystals at close-off. This latter parameter depends on the critical density of the snow, $D_0$, which corresponds to the transition between grain-boundary sliding (GBS) and power law creep (PLC) as the dominant densification mechanism: the higher the critical density of snow, the greater the number of pores per grain and therefore the larger TAC value at close-off (Arnaud, 1997). Since GBS decreases for larger grains (Alley, 1987), while PLC does not depend much on the grain size, $D_0$ should also decrease when the grains are big.

Thus, time periods with a warmer local summer temperature (due to high local insolation) promote a coarser-grained snow structure and hence lower critical density of snow and reduced TAC at pore closure, and vice versa. This mechanism is proposed here to explain the strong anti-correlation observed between TAC and the mean summer surface temperature. A numerical model, which takes into account the successive mechanisms involved between the surface snow and the closure of pores, is still required.

**7 Conclusions**

The lack of seasonal temperature reconstruction on Antarctica hampers a good understanding of the forcing and mechanism of climate changes over this climatically sensitive region. In this study, we revisit the TAC record

measured in the EDC ice core covering the last 440 ka. We show that it is dominated by a 40-kyr periodicity and is anti-correlated with the local mean insolation over the astronomical half-year summer. In order to investigate further this link between local summer insolation changes and TAC variations, we look into the correlation between the EDC TAC record and simulated local summer temperature changes by the LOVECLIM1.3 model. We evidence also an anti-correlation between those two independent variables. We explain the anti-correlations between local summer insolation/temperature and the EDC TAC by proposing that (1) the local summer insolation is controlling the development of strong temperature gradients in the near surface snow during the summer, (2) those summer temperature gradients are then modifying the surface snow structure and eventually (3) these snow structure changes propagate through the firn during the densification process down to the close-off depth where they impact the pore volume, i.e. the TAC of air bubbles (Lipenkov et al., 2011; Raynaud et al., 2007). These results points towards the fact that the EDC TAC record could be used as a unique proxy for local summer temperature. Future studies should investigate this relationship between TAC variations and local summer temperature changes in other ice core records drilled in Antarctica and Greenland.

The comparison between TAC and δD records indicates that the major driving factors for the summer and annual mean temperatures are different at EDC. TAC is strongly linked to local summer temperature, while the annual mean temperature is strongly controlled by the glacial-interglacial boundary conditions like the global ice volume and GHG. We show that the LOVECLIM1.3 model could capture the orbital-scale variability of the δD-based temperature reconstruction. Our transient simulation which allows us to investigate the relative effect of insolation, atmospheric greenhouse gas concentrations and NH ice sheet volume changes, shows that as compared to insolation, GHG and NH ice sheets have weak effect on the summer temperature, but strong effect on annual mean temperatures. Future modeling studies should investigate also the impact of past Antarctic ice sheet changes on local summer temperatures and consequently on TAC records. Overall, our model results confirm the hypothesis made from the spectral characteristics of the EDC TAC and δD records, explaining why these two records display different orbital periodicities.

### Author contributions

DR and QY designed the research. QY, ZW and AB performed the research related to the LOVECLIM1.3 simulations. DR, FP, EC and VL performed the research based on the EDC TAC record. QY, EC and DR wrote a draft of the paper with subsequent inputs from all the other authors.

**Competing interests**

At least one of the (co-)authors is a member of the editorial board of Climate of the Past.

**Acknowledgments**

The authors would like to warmly thank Jean Jouzel and Paul Duval for insightful discussions. The authors are also grateful to the two anonymous reviewers and Edward Brook for their constructive comments that helped improving this study. The modelling part of this work was supported by the Fonds de la Recherche Scientifique-FNRS (F.R.S-FNRS) under Grants n° T.0246.23 and T.W019.23. QY is Research Associate F.R.S.-FNRS. ZW is supported by the F.R.S.-FNRS Grant T.0246.23. Computational resources have been provided by the supercomputing facilities of the Université catholique de Louvain (CISM/UCL) and the Consortium des Équipements de Calcul Intensif en Fédération Wallonie Bruxelles (CÉCI) funded by the Fond de la Recherche Scientifique de Belgique (F.R.S.-FNRS) under convention 2.5020.11. The research leading to these results has received funding from the French National Research Agency under "Programme d'Investissements d'Avenir" (ANR-19-MPGA-0001) through the Make Our Planet Great Again HOTCLIM project. EC also acknowledges the financial support from the AXA Research Fund. This work is a contribution to EPICA, a joint European Science Foundation/European Commission (EU) scientific program, funded by the EU and by national contributions from Belgium, Denmark, France, Germany, Italy, The Netherlands, Norway, Sweden, Switzerland, and the UK. This is EPICA publication number X.

**Data availability:** The data used in this study will be uploaded to https://zenodo.org once the paper is accepted for publication.

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
