# Peer review of "Local summer temperature changes over the past 440 ka revealed by the Total Air Content in the Antarctic EPICA Dome C ice core"

_EGUsphere, 2023_

## Author Comment (AC1)

**Reply to Review 1**

We thank Reviewer 1 for the constructive and insightful comments. Here is our point-by-point reply. The review comments are in black and our replies are in red.

The authors use existing ice core data and new climate model simulations to provide a new interpretation of the ice core total air content (TAC) proxy. Based on a model-data correlation analysis, they suggest that the EDC TAC record reflects local summer temperature. The idea is interesting if corroborated, and the paper is overall well-written.

**Main comments:**

(1) After reading the paper, it was not exactly clear to me what the authors believe to be the mechanism for the TAC-summer temperature relationship. Which of these is it:

(a) Insolation controls summer temperature. Summer temperature controls TAC.
(b) Insolation controls summer temperature. Insolation controls TAC.
(c) We cannot distinguish between scenarios (a) and (b) based on the available evidence
(d) Something else altogether.

Please indicate which of these options describes your understanding best, and please clarify this in the manuscript as other readers may have the same confusion as I do.
We mean option (a). The link between insolation and TAC is through summer temperature. In the introduction (p.2, lines 59-62) of our manuscript we have explained that the anti-correlation between local summer insolation and TAC can be attributed to a mechanism where the local summer insolation is controlling the near-surface snow temperature and temperature gradients during summer time, which affects the near surface snow structure and hence TAC. In section 6 we discuss the possible mechanisms linking TAC and local summer temperature. We propose a mechanism based on snow/firn physics, which could explain the strong anti-correlation observed between TAC and the mean summer surface temperature. Nevertheless, a numerical model, which takes into- account the successive mechanisms involved between the surface snow and the closure of pores, is still required. Such model would explain that time periods with higher summer insolation and summer temperature will promote a coarser-grained snow structure, a lower critical density of snow and then a reduced TAC at pore closure.
 We will stress more about this point in the revised manuscript.

 (2) The analysis relies on correlation analysis between the EDC TAC record, the mean insolation over the local astronomical half-year summer, and the modeled summer temperature. The correlation between TAC and modeled summer T is stronger than the correlation between TAC and insolation alone. Could this be because both the TAC and summer temperature have 100ka power, whereas the insolation metric does not? Presumably the summer T has 100ka power because the GHG forcing that was applied.

I would request the authors include additional figures to clarify these questions better.
In Figure 1 could you please add the time series and spectrum also for the (i) mean insolation over the local astronomical half-year summer, (ii) modeled summer temperature, (iii) modeled mean-annual temperature? This could also be a separate figure of course.
Considering the suggestion of the reviewer, we will add a separate figure showing the time series and spectrum for the mean insolation over the local astronomical half-year summer insolation and the modeled summer temperature (see Figure R1 below). We are not in favor to add a wavelet/transform figure for the simulated annual mean temperature from the 10x accelerated simulation because the annual mean temperature is more strongly affected by ice volume (which is not the case for the mean summer temperature as explained in Section 5), but ice sheet changes were not considered in our 10x accelerated simulation, so a wavelet/spectrum figure of this simulated annual temperature would be misleading. However, the time series of the simulated annual

mean temperature is shown in Figure R3 and this figure will also be included in the revised version of the manuscript.

[Figure]

Figure R1. Time series and spectrum for (a) the mean insolation over the local astronomical half-year summer and (b) the modeled summer temperature.

In Figure 2 could you please add a panel with the correlation between Half-year summer insolation and the modeled summer T

Yes, we will add in Figure 2 a panel (see Figure R2 below) with the timeseries and correlation of mean summer insolation and simulated summer temperature.

[Figure]

Figure R2. (a) Raw and filtered timeseries of the half-year summer insolation and the half-year summer temperature, (b) correlation between the two filtered timeseries.

We will also add a panel in Fig. 4 to show the wavelet analysis of the mean summer insolation (see Figure R3 below). The reviewer is right in the fact that there is no 100-ka cycle in the mean summer insolation at 75S (Figure R3c), and the simulated summer temperature is driven by both insolation and GHG although insolation plays a dominant role. This is why the 100-ka cycle is still apparent in the simulated summer temperature due to GHG forcing.

[Figure]

Figure R3 (updated Figure 4). Continuous wavelet transforms of (a) the low-pass (> 12 ka) filtered TAC record, (b) simulated mean half-year summer temperature at the EDC site and (c) mean insolation during astronomical half-year summer at 75°S.

(3) Could you provide a figure that compares the accelerated simulations with the non-accelerated simulations for both summer and annual-mean temperatures? It seems important to assess how well these agree with one another.

We will add Figure R4 below in the revised manuscript to show the good agreement between the accelerated and non-accelerated simulations for both summer and annual-mean temperatures.

[Figure]

Figure R4. Comparison of 10x accelerated (purple) and non-accelerated (blue) simulations for the simulated mean half-year summer (October to March) temperature (upper panel) and mean annual temperature (lower panel) at EDC. Both are OrbGHG simulations without considering ice sheet changes. 1000-year running mean is plotted.

(4) How well does the model succeed in simulation the modern-day EDC annual temperature cycle? That may be important in assessing how well it can simulate the seasonal contrast back in time. Does the model simulate the Antarctic inversion, or is this not resolved in an EMIC? My sense is that the success of the model in simulating the seasonal temperatures will be closely linked to how well it can simulate the strong Antarctic

inversion. The inversion is likely to respond in different ways to ORB, GHG and ICE forcings, which may bias the model output. Please add some discussion and/or caveats to the paper on this aspect of the modeling. Figure R5 below shows the comparison of the modern climate in Antarctica simulated by LOVECLIM1.3 with observation. LOVECLIM1.3 reproduces quite well the spatial pattern and the magnitude of surface temperature over Antarctica in winter, summer and annual mean (Figure R5a). It is slightly cooler in western Antarctica in the model probably related to its low resolution. The seasonal temperature cycle at the EDC site is also well reproduced by the model (Figure R5b). The model also simulates the Antarctic inversion (Figure R6). To address the comment of the reviewer on the potential impact of Orb, GHG and ICE forcings on the inversion, the mode results at a few selected dates are plotted in Figure R6. It shows that these forcings can slightly affect the vertical temperature distribution, but they do not alter the inversion. Figures R5-6 and related discussions will be added in the revised manuscript.

(a)                                                      (b)

[Figure]

Figure R5. Comparison of the 1971-2000 mean climate in Antarctica simulated by LOVECLIM1.3 and the ERA5 reanalysis (https://cds.climate.copernicus.eu/cdsapp#!/home). (a) Surface temperature for the mean annual, summer (DJF) and winter (JJA); (b) The seasonal cycle of the surface temperature at the EDC site.

[Figure]

Figure R6: Temperature profile simulated by LOVECLIM 1.3 for annual mean. From left to right: the 1971-2000 mean; effect of obliquity, with small obliquity at 114ka (22.38°) and large obliquity at 94 ka (24.256°), both having similar precession; effect of GHG, with $CO_2$ being ~70 ppmv less in OrbGHG_88ka than in Orb_88ka; and effect of ice sheets at 91 ka, with the NH ice volume being ~20 million $km^3$ larger in OrbGHGIce_91ka than in OrbGHG_91ka. TS means surface.

**Minor comments:**

Line 20: "summer insolation" is vague in this context. Do you mean "mean insolation over the local astronomical half-year summer" again, or something else?

Yes, it means "mean insolation over the local astronomical half-year summer". To avoid confusion, at line 20, we will change "local summer insolation" to "mean insolation over the local astronomical half-year summer". In the main text, to avoid frequent repetition of such a long sentence, we use "mean summer insolation" to represent "mean insolation over the local astronomical half-year summer", as stated in line 137 in our original manuscript.

Line 25-26: "Mean annual temperature": technically it would be the "precipitation-weighted condensation temperature"
We would like to keep the beginning of the introduction more generic and less technical for a better readability for most readers, so we suggest to use "Mean annual temperature" in Line 25-26, but in line …. at the beginning of section 5, we will mention that it is "precipitation-weighted condensation temperature".

Line 35: I think Southern Ocean sea ice extent is another key parameter here
Thanks for the suggestion. Southern Ocean sea ice extent will be included in this sentence.

Line 83: and "a" detailed description….
Yes, corrected.

Line 91-98: why not change SH ice sheets? Those would be the most important for EDC, no?
It would be ideal to change also SH ice sheets, but there is a lack of well reconstructed/simulated geometry of the SH ice sheets for our study period that could be easily prescribed in our model. We agree that it would be interesting to test the impact of the Antarctica ice sheet change on the EDC TAC record in future studies. We will add that idea in the perspectives of the revised manuscript.

Figure 1B: the magnitude of EDC glacial-interglacial temperature change is uncertain, as per the discussion in the paper later on. It may be better to just plot d2H or d18O instead.
We will change it to the EDC δD record in the revised version. It does not change the spectral characteristic of this proxy.

Line 114 -115: I understand what is meant here, but the language is imprecise as there are not different "types" on insolation. Maybe rephrase to: "Comparing a proxy record with insolation is not necessarily straightforward because different metrics of insolation exist and their relationship with climate is not always clear"
Thanks for the proposition. It will be changed to metrics.

Line 126: perhaps remind the reader that 75 degrees S is the latitude of EDC.
Yes, the sentence will be changed to "the mean insolation over the astronomical half-year summer at 75°S (the latitude of EDC).

Figure 2: Please add a third row of panels in which you compare the half-year summer insolation to the modeled half-year summer temperature. That way the reader can evaluate the relationship between these two key parameters better
Such a third row will be added.

Line 172-175: is it possible to evaluate how well LOVECLIM simulates the cited reconstructed seasonal temperatures from West Antarctica to validate the model simulations?
In Figure R7 below we plotted our simulated summer and winter temperatures at the WDC site against Fig.2 of Jones et al 2022. Our simulated summer temperature compares well with the reconstructed summer temperature in both the trend and the magnitude of temperature change. Both show that summer temperature in west Antarctica had an increasing trend from early to mid-Holocene and reached a maximum at ~4 ka BP followed by a decreasing trend. Our simulated winter temperature is quite different from the reconstructed one especially in the early Holocene during which our model simulates a cooling trend like in the ORBIT simulation of HadCM3, but the reconstruction shows a weak warming trend. According to the GLAC1D and ICE-6G simulations of HadCM3, using different ice sheet configurations could significantly influence the

simulated winter temperature at WDC. Another possible reason for the mismatch between our simulated winter temperature and the reconstruction is that our model has a relatively low sensitivity in response to $CO_2$ change so the simulated warming due to a small $CO_2$ increase in the early Holocene is too weak in our model, letting the orbital forcing dominate.

A brief introduction of this comparison will be added in our revised manuscript.

[Figure]

Figure R7. Top and bottom panels: Simulated summer and winter temperature anomalies at WDC by LOVECLIM 1.3 OrbGHGIce transient simulation; Two middle panels: Fig. 2 of Jones et al. 2022.

Line 181-182: Please provide more justification for this statement. Is this based solely on the increased correlation coefficient?

The linear coefficient between TAC (filtered) and simulated summer temperature indicates that about 58% of the TAC variability observed at EDC over the last 440 ka is explained by the half-year summer temperature. Nevertheless, this statement is based solely on this correlation coefficient, but there are more factors affecting TAC, as mentioned in the paper. Please see also more explanation in our reply to main comment 1.

Line 186: "This is information…." (information is singular)
Done.

Line 185-191: Can you clarify this discussion? Is TAC just a proxy for summer temperature, or is summer temperature actually driving TAC variations? The discussion here still assumes that TAC is driven by insolation directly.
We do not mean that TAC is driven by insolation directly. As explained in our reply to main comment 1, we mean that local summer insolation controls the temperature and the vertical temperature gradient in near-surface snow. Then, the surface snow structure is physically affected by changes in summer temperature. This surface structure change driven by summer temperature controls TAC. So, TAC can be used as a proxy for summer temperature. Furthermore, more detail on the physical mechanisms is explained in section 6. Our view is: insolation controls summer temperature, and summer temperature controls TAC, so TAC can be used as a proxy for summer temperature.
This discussion will be clarified and elements will be added in the revised manuscript.

Figure 5: For comparison, can you please also plot the mean annual temperatures from the accelerated simulations that span the full 440ka period?
As mentioned in our reply to main comment 3, a new figure will be added on the comparison of the annual mean temperature between the accelerated and non-accelerated simulations.

Line 217: "Response of Antarctic climate" (remove the 'a' at the end of Antarctica)
Yes, corrected.

Line 221-222: In light of the discussion that follows, perhaps rephrase in more neutral language? For example: "One may also note that the magnitude of the temperature change between glacial and interglacial is significantly smaller in the model as compared to the reconstruction"
OK, we propose to rephrase in the revised manuscript the sentence such as "One may also note that the magnitude of the temperature change between glacial and interglacial is significantly underestimated in the model as compared to the reconstruction".

Line 223: It is my understanding that Antarctic elevation changes would exacerbate the problem, as EDC likely had lower elevation during the LGM.
It is an interesting point that deserves to be further studied. In principle, during glacial periods, the reduced surface accumulation rate leads to lower surface elevation (see Figure R8 below taken from the appendix of Raynaud et al., 2007). But there are also dynamical effects which make these reconstructions of surface elevation uncertain. Surface elevation changes affect our study in two different ways. First, TAC should be corrected for atmospheric pressure changes to get a record of porosity at close-off. Some of these atmospheric pressure changes are due to variations in surface elevation, another part might be due to change of atmospheric conditions (like the temperature of the air column). Because the reconstruction of surface elevation changes are uncertain, we have chosen to not correct for this effect. Second, surface elevation changes should also be taken into account in our simulation of summer or annual temperature at EDC by LOVECLIM. Unfortunately, in our climate model, the Antarctic temperature is fixed and we did not account for surface elevation changes. We could apply an a posteriori correction for surface elevation changes, but because the temperature variations (either annual of in summer) are probably under-estimated, applying this a posteriori correction would have a too strong influence. Finally, we should note that these two elevation corrections (for TAC and for the climate model) go in the same direction: during glacials, a corrected summer temperature would be warmer, and the corrected TAC of the ice would be smaller, so there is a chance that these two corrections would cancel each other and that the overall correlation between TAC and modeled summer temperature would not be much affected.
We will include this discussion in the corrected manuscript.

[Figure]

Figure R8. Simulated elevation changes at EDC from Raynaud et al. 2007.

Line 239-251: This is really excellent, and very instructive. What is the role of winter temperature? Is it fully controlled by GHG, such that in the annual average both GHG (winter) and ORB (summer) show up?

Thanks for this inspiring question. The wavelet analysis of the simulated mean half-year winter temperature at EDC (Figure R9 below) show very strong ~20-kyr cycle and an obvious ~100 kyr cycle, but the ~40-kyr cycle is very weak. The 100-kyr cycle results most probably from the effect of GHG. The very strong 20-kyr cycle but very weak 40-kyr cycle is quite intriguing. As far as insolation is concerned, the low-latitude insolation is dominated by the ~20-kyr precession cycle. As the solar energy received in Antarctica is very weak during local winter, the strong 20-kyr cycle in the simulated winter temperature could reflect a strong effect of the low latitude climate on the Antarctica temperature during austral winter, via for example meridional ocean and atmosphere heat transport. Figure R9b shows a high, negative correlation between the simulated EDC winter temperature and precession. This indicates that the EDC winter temperature is strongly affected by boreal summer insolation in low latitudes (small precession parameter leads to high boreal summer insolation and vice versa). It is also shown in Yin and Berger (2012) that during some interglacials such as MIS-5e, -15 and -17 which are characterized by very strong boreal summer insolation, a strong warming could be induced over Antarctica during austral winter, a warming which is even stronger than in many other regions due to polar amplification.

[Figure]

Figure R9: (a) Continuous wavelet transform of the simulated mean half-year winter temperature at the EDC site from the 10x accelerated OrbGHG simulation, and (b) Correlation between this winter temperature and precession.

In our manuscript, the relative effects of insolation, GHG and NH ice sheets on the summer and annual temperature at EDC have been studied using the Orb, OrbGHG and OrbGHGIce experiments for the period 133-75 ka. The same experiments are used here to analyze their effects on the winter temperature. Figure R10 below shows that similar to what happens to the summer temperature, orbital forcing plays a dominant role

also on the winter temperature at EDC. As explained above, the winter temperature at EDC is actually strongly driven by precession and boreal summer insolation, so on precession timescale, the orbitally-induced temperature variation in winter is in anti-phase with the summer temperature which is strongly driven by austral summer insolation. This anti-phase relationship leads to a strong weakening of the orbital signal especially the precession signal in the mean annual temperature (see Figure R10 below), making the effect of GHG and ice sheets more pronounced and leading to strong glacial cycles in the mean annual temperature.

[Figure]

Figure R10: (a) Mean half-year summer (October to March) temperature, (b) mean half-year winter (April-September) temperature and (c) mean annual temperature at the EDC site from the LOVECLIM1.3 transient simulations without acceleration for the period 133-75 ka. Black curves are the results from the Orb simulation under only orbital forcing; blue curves are the difference between OrbGHGIce and Orb simulations, showing the joint effect of GHG and ice sheets.

These discussions and figure will be added in the revised manuscript.

Line 249: The GHG effect is 1 degree on both mean-annual and summer temperatures, so it is actually the same. Perhaps better to phrase it as the *relative* importance of GHG is bigger on annual-mean.
Thanks for the suggestion. It is indeed better to use "relative". The sentence will be changed to "These clearly show that as compared to insolation, GHG and NH ice sheets have relatively weaker effect on the summer temperature but they have relatively stronger effect on annual mean temperature".

Fig, 6: The model appears to simulate a seasonal temperature cycle of around 16 degrees (twice the offset between summer and annual-mean). The observed seasonal cycle is closer to 40 degrees (from -25 in summer to -65 in winter). Please comment. How well does the model capture the seasonal variations in the modern day?

Please see our reply to main comment 4. The model captures quite well the seasonal temperature variations at EDC. New text and figures will be added in section 2.2 to comment on the performance of the model for the modern seasonal temperature cycle at EDC.

Section 6: can you me more clear in your usage of the words snow and firn? How do you define these? Except for the very fresh surface snow, anything older than 1 year I would call firn. But it appears you use the terms differently.

Indeed, there are no generally accepted definitions of snow and firn in the literature, and often these two terms are used interchangeably. Some papers on snow/firn metamorphism use only the term "firn" (e.g. Alley, 1987) while others, conversely, use only "snow" (e.g. Maeno & Ebinuma, 1983). The definition adhered to by the reviewer is more commonly found in the American literature (see, for example, Cuffey & Paterson, 2010, The Physics of Glaciers). However, in our work we follow the definition introduced by Anderson and Benson (1963), who associate snow and firn (névé in the original) with two different stages in the transformation of dry snow into bubbly ice. These two stages differ in the mechanisms that dominate the densification process: the particle rearrangement controlled by linear-viscous grain-boundary sliding (GBS) dominates in snow, whereas the power-law creep (PLS) dominates in further densification of firn. The snow-firn transition, viewed as a transition between the two densification regimes, is identified by the bending of the density-depth profile, which indicates a decrease in compaction rate and is usually observed when the critical snow density, equal to about 550 kg/m3 (which corresponds to a relative density of $D_0 = 0.6$ and is not a constant!), is reached. At this density, the number of contacts per grain approaches 6–7, thus making the sliding impossible. "Recognition of the critical density introduces the possibility of making a physical distinction between snow and firn, or névé" (Anderson & Benson, 1963). This conceptual framework has been adopted in a number of papers dealing with the physical modeling of snow-to-ice transformation (e.g. Arnaud et al, 1998, 2000) and the results of these works have been used in (Raynaud et al, 2007; Lipenkov et al, 2011) as a basis for discussing a possible mechanism of the effect of summer temperatures on the pore volume of firn at the close-off depth. We will clarify the usage of these two words in the revised manuscript.

Line 273: "ice grains" instead of "snow grains"?
Yes, ice grains would be the correct wording.

Line 274: "upper few meters of THE snow column"
Done.

Line 278: the critical density you refer to here is around 550 kg/m3, correct? Or do you mean the critical density for pore closure? Please specify for clarity. What depth is this at EDC?
Yes, the first is correct. We also felt that the whole sentence was a bit misleading. To make the text clearer, we will change it as follows in the revised manuscript:
*"According to the model proposed by Arnaud (1997) for the Antarctic ice sheet, the pore volume at close-off, Vc, should increase with the mean annual surface temperature through the competing densification mechanisms: higher temperature leads to an increase in the relative critical density at the transition between snow and firn ($D_0$~0.6, reached at a depth of about 25m at EDC), which in turn implies a greater proportion of the ice-grain edges occupied by pores at close-off, and hence a larger Vc."*

Line 279-280: is it no correlation or a poor correlation? Has to be either, can't be both.
It is poor correlation. It will be changed in the revised manuscript.

Line 281: there is no surface temperature record. Do you mean a correlation between TAC and the modeled summer temperature
Yes, we mean a correlation between TAC and the simulated mean summer surface temperature. It will be modified.

Line 283: what is the number of pores per grain? Simply the ratio of the nr. of pores to the number of ice crystals?
Yes. We will define this term in the revised version.

Line 284: do you mean the critical density of firn?
The critical density at the transition between snow and firn. Please see our replies above.

Line 284: This transition between GBS and PLC occurs at around 550 kg/m3, correct? Please specify for clarity. What depth is this at EDC?
Yes, this is correct. Please see our explanations and replies above.

Line 304-307: See my main comment, I would appreciate more clarity on what the authors think the causal relationships between insolation, summer temperature, and TAC are.
The causal relationships between insolation, summer temperature and TAC could be explained by that (1) local summer insolation is controlling the near surface summer temperature conditions, (2) this summer temperature conditions affect the near surface snow structure and then (3) the continuous nature of snow transformation until the trapping of the air in ice, i.e. TAC. So, the mechanism for the EDC TAC-summer temperature relationship is that insolation controls summer temperature and summer temperature controls TAC.

Line 308: Ultimately this insight does not come from the TAC record, but from the climate model. Do you agree? Without the climate model, one would not have interpreted TAC as a summer temperature proxy.
We don't totally agree. In Raynaud et al. 2007, TAC was already proposed to reflect summer temperature, and a reasonable physical mechanism could be proposed (also in the present manuscript) to explain how summer temperature could affect TAC. In this manuscript, the good comparison between TAC and the simulated summer temperature confirms the TAC could be a proxy for summer temperature. TAC itself has very different spectral characteristics from the δD which is used as an annual temperature proxy. So the statement of line 308 could be made based on both the TAC record and the model results.

**References:**

Alley, R., 1987. Firn densification by grain-boundary sliding: A first model. J. Phys. Colloques 48, DOI: 10.1051/jphyscol:1987135

Anderson, D.L. and G.S. Benson. 1963. The densification and diagenesis of snow: properties, processes and applications. In Kingery, VV. D., ed. Ice and snow: properties, processes, and applications. Cambridge, MA, M.LT.Press, 391- 41l.

Arnaud, L., V. Lipenkov, J.M. Barnola, M. Gay and P. Duval. 1998. Modelling of the densification of polar firn: characterization of the snow-firn transition. Ann. Glaciol., 26, 39-44.

Arnaud, L., J.-M. Barnola and P. Duval. 2000. Physical modeling of the densification of snow/firn and ice in the upper part of polar ice sheets. In Hondoh, T, ed., Physics of Ice Core Records, Hokkaido University Press, Sapporo, 285-305.

Lipenkov, V.Y., Raynaud, D., Loutre, M.F., Duval, P., 2011. On the potential of coupling air content and O2/N2 from trapped air for establishing an ice core chronology tuned on local insolation. Quaternary Science Reviews 30.

Maeno, N. and T. Ebinuma. 1983. Pressure sintering of ice and its implication to the densification of snow at polar glaciers and ice sheets. J. Phys. Chem., 87(21), 4103-4110.

Raynaud, D., Lipenkov, V., Lemieux-Dudon, B., Duval, P., Loutre, M.-F., Lhomme, N., 2007. The local insolation signature of air content in Antarctic ice. A new step toward an absolute dating of ice records. Earth Planetary Science Letters 261, 337-349, doi:310.1016/j.epsl.2007.1006.1025.

Yin, Q.Z. and Berger, A., 2012. Individual contribution of insolation and $CO_2$ to the interglacial climates of the past 800 000 years. Climate Dynamics, 38, 709-724.

---

## Author Comment (AC2)

**Reply to Review 2**

We thank Reviewer 2 for the constructive and insightful comments. Here is our point-by-point reply. The review comments are in black and our replies are in green.

**General comments:**
The manuscript explores the idea that there are no good reconstructions of summer temperature and suggests that TAC could be used as a proxy for summer temperature in Antarctica. The authors use a previously measured record of TAC from the EDC ice core, which spans the last 440 ka. They use the EDC TAC data to compare with a climate model simulation of summer temperature to make the case that TAC could be used as a summer temperature proxy.
Overall, the manuscript is good, and the idea for using TAC as a summer temperature proxy is important, especially if it could be shown to also be used on other ice cores (a subject for future research). I recommend its publications after the authors consider the below comments:

**Specific comments:**
Overall, it is difficult to follow if the authors main point is that the TAC is controlled by summer insolation or summer temperature, or both. The authors clearly describe that both summer insolation and summer temperature are well anti-correlated with TAC. To make this clearer, I would recommend reorganizing the conclusion, highlighting their main argument at the beginning of the first paragraph.
Reviewer 1 made the same remark. Here we use the same reply to Reviewer 1's main comment 1:
*We mean option (a). The link between insolation and TAC is through summer temperature. In the introduction (p.2, lines 59-62) of our manuscript we have explained that the anti-correlation between local summer insolation and TAC can be attributed to a mechanism where the local summer insolation is controlling the near-surface snow temperature and temperature gradients during summer time, which affects the near surface snow structure and hence TAC. In section 6 we discuss the possible mechanisms linking TAC and local summer temperature. We propose a mechanism based on snow/firn physics, which could explain the strong anti-correlation observed between TAC and the mean summer surface temperature. Nevertheless, a numerical model, which takes into- account the successive mechanisms involved between the surface snow and the closure of pores, is still required. Such model would explain that time periods with higher summer insolation and summer temperature will promote a coarser-grained snow structure, a lower critical density of snow and then a reduced TAC at pore closure.*

 We will stress more about this point in the revised manuscript and reorganize the conclusion accordingly.

Line 190 the authors state that TAC can be used as a proxy for summer temperature, based on the strong anti-correlation with modeled summer temperature. Then in section 6 the authors highlight that summer temperature influencing firn metamorphism is only a low-accumulation site phenomenon. Does this mean that TAC could not be used as a temperature proxy in Greenland? If this is the case, maybe the language about using TAC as a proxy for summer temperature is too strong for their results. While their result is interesting, before saying that TAC is a proxy for summer temperature, the relationships should be verified at multiple ice core sites.
Thank you for this important question. We should make it clear that our conclusion applies to the EPICA Dome C but that it has to be demonstrated for other sites in Antarctica and in Greenland. Note that this is an on-going work led by a PhD student at IGE. Going in this direction and to be more precise we will change the title of our paper into:
"Past local summer temperature changes revealed by the Total Air Content record from the EPICA Dome C ice core".

Line 190, we will also modify the sentence into: "The good correlation between the two independent climate variables, TAC measured from the EDC ice core, and local summer temperature simulated by a model, indicates that the EDC TAC record can be used as a proxy for local summer temperature. Future studies will

investigate this relationship between TAC variations and local summer temperature changes in other ice core records drilled in Antarctica and Greenland".

We will also mention this point in the conclusion of the revised manuscript.

Line 49: What is the difference between V (air content) and TAC (total air content)? A differentiation of what the authors mean by V vs TAC is required.
We wrote (lines 49-50):" *During previous works V (for air content) and TAC (for Total Air Content) have been indifferently used for designating the same property. In this work we are using TAC, which is usually used in the recent works.*"
So, there is no difference between the use of V and TAC, both designating the same property. As said, in this work we are using TAC, which is currently used in the recent works. We will make sure this is clear in the revised manuscript.

Line 84: What is a 10x acceleration, and why is it important in this context? I'm guessing the 10x model was used to save resources, but I recommend that more information is presented about why the accelerated models were used instead of the non-accelerated simulations.
Indeed the 10x acceleration was used to save computing resources and time. This technique can be explained by the text below cited from page 4 of Yin and Berger (2015) which is referred in our manuscript:
*"Although being a model of intermediate complexity, LOVECLIM remains still costly for transient experiments, particularly when 5 interglacials and 10 transient simulations are considered. An acceleration technique similar to Lorenz and Lohmann (2004) was therefore used to speed up the simulations and reduce the computational costs. An acceleration factor of 10 is used, which means that at the end of each year of the simulation, the astronomical parameters and GHG concentration are advanced by 10 years. In such a case, the actual length of the simulation is reduced by 10 times. For example, a 20,000-year long simulation only needs 2000 model years. To test the impact of such an acceleration technique on our transient simulations, a non-accelerated experiment and a 10-time accelerated one have been done for two interglacials MIS-5 and MIS-13. Our results showed that the acceleration method has little impact on the surface air temperature and precipitation. However, the response of the deep-ocean temperature is delayed by 2-3 ka in the accelerated simulations as compared to the non-accelerated ones, similar results being observed also in other studies (Timm and Timmermann, 2007; Ganopolski and Calov, 2011). A detailed analysis made by Timm and Timmermann (2007) shows that a 10-time acceleration leads to a delayed response of the temperature only in the inner ocean. As here we are mainly interested in surface climates, the 10-time acceleration technique would not alter our conclusion about the phasing between the surface temperatures of different regions."*

In our manuscript, the results of the 10x accelerated simulation are used for the entire 440 ka, but non-accelerated simulations for five glacial-interglacial episodes are also used to compare with the accelerated one. The comparison shows that the 10x acceleration technique does not affect the summer and annual temperature that were discussed in our manuscript. In our revised manuscript, we will give more information on these aspects.

Line 111: The sentence beginning "This spectral characteristic…" suggests that orbital astronomical forcing drives changes in TAC. The studies correlations do show a link between TAC and orbital patterns, but I think this sentence is misleading, and should be changed to reflect that TAC is only correlated with astronomical forcing, not actually caused by it. Later, I believe the authors make the argument that temperature gradients are what is causing TAC to vary, not insolation alone.
Thank you for this useful comment. We will rephrase the sentence in the revised manuscript. Also, as explained in our reply to referee 1, in the discussion on line 185-191 of the manuscript:
*"...we assume that local summer insolation controls the temperature and the vertical temperature gradient in near-surface snow, but not TAC directly. Then, the surface snow structure is physically affected by changes in summer temperature. This surface structure change driven by summer temperature will control TAC. So, TAC can be used as a proxy for summer temperature. Furthermore, more detail on the physical mechanisms is*

*explained in section 6. Our view is: insolation controls summer temperature, and summer temperature controls TAC, so TAC can be used as a proxy for summer temperature."*

Line 127: I like the description of the astronomical half-year, and it is intuitive as to why it would be used as opposed to ISI.
Thank you.

Line 151: Here the authors say that TAC can be considered a proxy driven by mean local summer insolation based on the correlation. I'm a bit unclear here, because the correlation is higher (0.58) between TAC and summer temperature than between TAC and insolation (0.39). Do the authors mean that insolation drives summer temperature drives TAC? Or are they referring to previous works results using integrated summer insolation? If so, what was the correlation coefficient in that instance?
Yes, we propose that TAC is driven by summer temperature which is in turn mainly driven by local summer insolation. However, as explained in our reply to Reviewer 1's main comment 2, although local summer insolation is a main factor controlling summer temperature, other factors such as GHG could also contribute. This is why the correlation between TAC and summer temperature is higher than the correlation between summer temperature and insolation.

Line 149, 150, 170: The authors use the term 'correlated' when I think they mean anti-correlated, or negatively correlated. This needs to be clear, as it can cause confusion. I recommend all instances are reviewed.
Yes, sorry for the confusion, we meant anti-correlated.

Line 177: Figure 2 does not show any correlation between summer temp and summer insolation. Recommend adding this to figure 2 or deleting this line.
A figure will be added in the revised manuscript.

Line 280: What does "no (poor) correlation" mean? Please give a correlation coefficient. You reference section 5, but maybe figure 1 would be a better reference?
We mean poor correlation. You are right that Figure 1 is a better reference. We will modify this line accordingly.

Line 282: What does it mean to 'affect negatively" the critical density? Does this mean the other factor decreases the critical density?
Yes, this sentence needs rewording. The modified text reads: "This observation suggests that summer temperature has an inverse effect on Vc compared to the mean annual temperature". It will be included in the revised manuscript.

Line 308: I thought the comparison came from dD and modelled summer temperature, not TAC and dD?
We do mean the comparison between TAC and δD. A simple comparison of their spectral characteristics already indicates that their major drivers are different.

**Technical Comments:**
 Line 3 – insert ",a" before "..tracer"
Done.

Multiple places EDC is referred to as Dome C. Recommend using EDC throughout the paper.
Done.

Line 155 – Recommend starting new paragraph here, where you start to discuss orbital tuning.
Done.

Figure 3 units – Are your units on the left hand side correct? I think they should be the same order of magnitude as the right side.

We are sorry, there is indeed a mistake in the units on this figure. It will be corrected in the revised manuscript.

Line 211 – Strange spacing issue here.
Indeed, there should be no space here. It will be changed.

Line 290: Recommend deleting the second 'here' in the sentence.
Done.

Line 312: Missing a word. Perhaps "Our transient simulation which allows us…"
Done.

**Reference:**

Yin, Q.Z. and Berger, A., 2015. Interglacial analogues of the Holocene and its natural near future. Quaternary Science Reviews 120, 28-46.

---

## Author Response (AR1)

First we would like to thank the editor for his additional comments that we have accounted for in our revised manuscript.

**Reply to Review 1**

We thank Reviewer 1 for the constructive and insightful comments. Here is our point-by-point reply. The review comments are in black and our replies are in red. Changes are indicated by yellow highlights in the revised manuscript version that indicates the modifications made according to the reviewer's comment.

The authors use existing ice core data and new climate model simulations to provide a new interpretation of the ice core total air content (TAC) proxy. Based on a model-data correlation analysis, they suggest that the EDC TAC record reflects local summer temperature. The idea is interesting if corroborated, and the paper is overall well-written.

**Main comments:**

(1) After reading the paper, it was not exactly clear to me what the authors believe to be the mechanism for the TAC-summer temperature relationship. Which of these is it:

(a) Insolation controls summer temperature. Summer temperature controls TAC.
(b) Insolation controls summer temperature. Insolation controls TAC.
(c) We cannot distinguish between scenarios (a) and (b) based on the available evidence
(d) Something else altogether.

Please indicate which of these options describes your understanding best, and please clarify this in the manuscript as other readers may have the same confusion as I do.
We mean option (a). The link between insolation and TAC is through summer temperature. In particular, we propose that the anti-correlation between local summer insolation and TAC can be attributed to a mechanism where the local summer insolation is controlling the near-surface snow temperature and temperature gradients during summer time, which affects the near surface snow structure and hence TAC. In section 6 we discuss the possible mechanisms linking TAC and local summer temperature. We propose a mechanism based on snow/firn physics, which could explain the strong anti-correlation observed between TAC and the mean summer surface temperature. Nevertheless, a numerical model, which takes into- account the successive mechanisms involved between the surface snow and the closure of pores, is still required. Such model would explain that time periods with higher summer insolation and summer temperature will promote a coarser-grained snow structure, a lower critical density of snow and then a reduced TAC at pore closure. We have tried and clarified our point in the revised manuscript in several places:

*In the abstract:*
*Line 21: We propose that (1) the local summer insolation controls the local summer temperature, (2) the latter, through the development of temperature gradients at the near surface of the ice sheet (<2m), is affecting the surface snow structure and (3) those snow structure changes propagating down to the bottom of the firn through densification are controlling eventually the pore volume at the bubble close-off and consequently, the TAC. Hence, our results suggest that the EDC TAC record could be used as a proxy for local summer temperature changes.*

*In section 4:*
*Line 244: Hence, we propose that the link between summer insolation and TAC variations exists through the summer temperature changes (see also section 6). Indeed as proposed in previous studies (Lipenkov et al., 2011; Raynaud et al., 2007), the local summer insolation controls the near-surface snow temperature and the vertical temperature gradients in snow. In turn, the latter affects the near-surface snow structure and consequently the porosity of the firn pores at close-off, i.e. the TAC.*

*In the conclusions:*

*Line 412: The lack of seasonal temperature reconstruction on Antarctica hampers a good understanding of the forcing and mechanism of climate changes over this climatically sensitive region. In this study, we revisit the TAC record measured in the EDC ice core covering the last 440 ka. We show that it is dominated by a 40-kyr periodicity and is anti-correlated with the local mean insolation over the astronomical half-year summer. In order to investigate further this link between local summer insolation changes and TAC variations, we look into the correlation between the EDC TAC record and simulated local summer temperature changes by the LOVECLIM1.3 model. We evidence also an anti-correlation between those two independent variables. We explain the anti-correlations between local summer insolation/temperature and the EDC TAC by proposing that (1) the local summer insolation is controlling the development of strong temperature gradients in the near surface snow during the summer, (2) those summer temperature gradients are then modifying the surface snow structure and eventually (3) these snow structure changes propagate through the firn during the densification process down to the close-off depth where they impact the pore volume, , i.e. the TAC of air bubbles (Lipenkov et al., 2011; Raynaud et al., 2007). These results points towards the fact that the EDC TAC record could be used as a unique proxy for local summer temperature. Future studies should investigate this relationship between TAC variations and local summer temperature changes in other ice core records drilled in Antarctica and Greenland.*

(2) The analysis relies on correlation analysis between the EDC TAC record, the mean insolation over the local astronomical half-year summer, and the modeled summer temperature. The correlation between TAC and modeled summer T is stronger than the correlation between TAC and insolation alone. Could this be because both the TAC and summer temperature have 100ka power, whereas the insolation metric does not? Presumably the summer T has 100ka power because the GHG forcing that was applied.

I would request the authors include additional figures to clarify these questions better.
In Figure 1 could you please add the time series and spectrum also for the (i) mean insolation over the local astronomical half-year summer, (ii) modeled summer temperature, (iii) modeled mean-annual temperature? This could also be a separate figure of course.
In Figure 2 could you please add a panel with the correlation between Half-year summer insolation and the modeled summer T

We have now added in the revised Figure 3 a panel (see below) with the time series and correlation of mean summer insolation and simulated summer temperature.
We have also added a panel in the revised Figure 5 to show the wavelet analysis of the mean summer insolation (see below). The reviewer is right in the fact that there is no 100-ka cycle in the mean summer insolation at 75S (revised Figure 5c), and the simulated summer temperature is driven by both insolation and GHG although insolation plays a dominant role. This is why the 100-ka cycle is still apparent in the simulated summer temperature due to GHG forcing.
Because of these changes on revised Figures 3 and 5, we propose to not add a panel in Figure 1 with the time series and spectrum for the mean insolation over the local astronomical half-year summer and the modeled summer temperature as we believe this would be redundant with the information already provided in those two revised figures and there are already a lot of figures in our revised manuscript.
Finally, we are not in favor to add a wavelet/transform figure for the simulated annual mean temperature from the 10x accelerated simulation because the annual mean temperature is more strongly affected by ice volume (which is not the case for the mean summer temperature as explained in Section 5), but ice sheet changes were not considered in our 10x accelerated simulation, so a wavelet/spectrum figure of this simulated annual temperature would be misleading. However, the time series of the simulated annual mean temperature is shown in now shown in the Figure 6 included in the revised version of the manuscript.

[Figure]

*Revised Figure 3. Comparison of TAC record with (a) mean insolation during astronomical half-year summer at 75°S calculated using the solution of Berger and Loutre (1991) and with (c) simulated mean half-year summer (October to March) temperature at EDC. Their corresponding linear regression analyses are shown in (d) and (d). (e) Comparison of half-year summer insolation with simulated mean half-year summer temperature and corresponding linear regression in (f). Low-pass filtered >12 kyr is applied on the TAC, insolation and summer temperature raw data before comparison. Note that the TAC axis is reversed on the left panels to ease the visual comparison.*

[Figure]

*Revised Figure 4. Continuous wavelet transforms of (a) the low-pass (> 12 ka) filtered TAC record, (b) simulated mean half-year summer temperature at the EDC site and (c) mean insolation during astronomical half-year summer at 75°S.*

[Figure]

*Revised Figure 6. Comparison of the EDC mean annual temperature record (pink line, Jouzel et al. 2007) with the simulated annual mean temperature of the OrbGHGIce experiments (blue line, this study).*

 (3) Could you provide a figure that compares the accelerated simulations with the non-accelerated simulations for both summer and annual-mean temperatures? It seems important to assess how well these agree with one another.

We have added in a supplementary material such a figure (Figure S1) to show the good agreement between the accelerated and non-accelerated simulations for both summer and annual-mean temperatures. We also discuss this figure in the revised manuscript.

*Line 109: Although LOVECLIM1.3 is classified as an EMIC model, its complexity is high for this kind of models and its ocean component is a full general circulation model, so it remains challenging to run full transient simulations with this model. We therefore first performed a transient simulation with 10x acceleration covering the last 800 ka, which allows to compare the simulated local summer temperature with the TAC record over the entire last 440 ka. In this simulation, the variations of orbital forcing and GHG were considered, and the global ice sheets were fixed to their pre-industrial condition. Using the same model and the same acceleration technique, it has been shown that 10x acceleration has a significant impact on deep ocean temperature, but it has no major impact on surface temperature (Yin and Berger, 2015). This is further confirmed in our study where the Antarctic summer and annual mean temperature changes of the 10x acceleration simulation are matching well with that of the non-accelerated simulations (Supplementary Fig. S1).*

[Figure]

*Figure S1. Comparison of 10x accelerated (purple) and non-accelerated (blue) simulations for the simulated mean half-year summer (October to March) temperature (upper panel) and mean annual temperature (lower panel) at EDC. Both are OrbGHG simulations without considering ice sheet changes. 1000-year running mean is plotted.*

 (4) How well does the model succeed in simulation the modern-day EDC annual temperature cycle? That may be important in assessing how well it can simulate the seasonal contrast back in time. Does the model simulate the Antarctic inversion, or is this not resolved in an EMIC? My sense is that the success of the model in simulating the seasonal temperatures will be closely linked to how well it can simulate the strong Antarctic inversion. The inversion is likely to respond in different ways to ORB, GHG and ICE forcings, which may bias the model output. Please add some discussion and/or caveats to the paper on this aspect of the modeling.

We have now added a new figure (revised Figure 2) in the revised manuscript to show the comparison of the modern climate in Antarctica simulated by LOVECLIM1.3 with observation. LOVECLIM1.3 reproduces quite well the spatial pattern and the magnitude of surface temperature over Antarctica in winter, summer and annual mean (Revised Figure 2a.). It is slightly cooler in western Antarctica in the model probably related to its low resolution. The seasonal temperature cycle at the EDC site is also well reproduced by the model (Revised Figure 2b). The model also simulates the Antarctic inversion (Revised Figure 2c). Note that the impact of Orb, GHG and ICE forcings on the inversion at different selected dates only slightly affect the vertical temperature, and they do not alter the inversion (not shown).

In the revised version of the manuscript, we also added:

*Line 104: In terms of the Antarctic climate, LOVECLIM1.3 reproduces reasonably the spatial pattern and the magnitude of surface temperature over Antarctica in winter, summer and annual mean (Fig. 2a). It is slightly cooler in western Antarctica in the model probably related to its rough resolution. The seasonal temperature cycle at the EDC site (Fig. 2b) as well as the Antarctic inversion (Fig. 2c) are also well reproduced by the model.*

[Figure]

*Figure 2. Comparison of the 1971-2000 mean climate in Antarctica simulated by LOVECLIM1.3 and the ERA5 reanalysis (https://cds.climate.copernicus.eu/cdsapp#!/home) for mean annual, summer (DJF) and winter (JJA) surface temperature (a) and for the seasonal cycle of the surface temperature at the EDC site (b), as well as the simulated vertical mean annual surface temperature profile at the EDC site.*

**Minor comments:**

Line 20: "summer insolation" is vague in this context. Do you mean "mean insolation over the local astronomical half-year summer" again, or something else?
Yes, it means "mean insolation over the local astronomical half-year summer". To avoid confusion, we have now replace the term "local summer insolation" by "mean insolation over the local astronomical summer half-year". In the main text, to avoid frequent repetition of such a long sentence, we use "mean summer insolation" to represent "mean insolation over the local astronomical summer half-year", as stated in line 179 in our revised manuscript.

*Line 21: "Our results show that the TAC record is highly anti-correlated with the mean insolation over the local astronomical summer half-year. They also show for the first time that it is highly anti-correlated with local summer temperature simulated with an Earth system model of intermediate complexity. This suggests that the Antarctic TAC records could be used as a proxy for local summer temperature changes. Also, our simulations show that the mean insolation over the local astronomical summer half-year is the primary driver of Antarctic summer surface temperature variations while changes in atmospheric greenhouse gas concentrations and northern hemisphere ice sheet configurations play a more important role on Antarctic annual surface temperature changes".*

Line 25-26: "Mean annual temperature": technically it would be the "precipitation-weighted condensation temperature"
We would like to keep the beginning of the introduction more generic and less technical for a better readability for most readers, so we suggest to use "Mean annual temperature" in line 20 of the revised manuscript. However, at the beginning of section 5, we now mention that it is "precipitation-weighted condensation temperature":

*Line 287: The δD record from Antarctic ice has been widely used as an important proxy for mean annual precipitation-weighted condensation temperature over Antarctica (Jouzel et al., 2007; Stenni et al., 2010), although it has also been suggested to be biased toward winter temperatures (Laepple et al., 2011).*

Line 35: I think Southern Ocean sea ice extent is another key parameter here
Thanks for the suggestion. Southern Ocean sea ice extent is now included in this sentence.
*Line 46: As the insolation over Antarctica is received mostly during summer, having a proxy of summer temperature would therefore be essential for helping to decipher the role of NH versus local insolation as well*

*as the role of other glacial boundary conditions such as the changes in atmospheric GHG concentrations, in Southern Ocean sea ice extent and in ice sheet configuration.*

Line 83: and "a" detailed description....
Done

Line 91-98: why not change SH ice sheets? Those would be the most important for EDC, no?
It would be ideal to change also SH ice sheets, but there is a lack of well reconstructed/simulated geometry of the SH ice sheets for our studied period that could be easily prescribed in our model. We agree that it would be interesting to test the impact of the Antarctica ice sheet change on the EDC TAC record in future studies. We have now added this idea in the perspectives of the revised manuscript.

*Line 434: Future modeling studies should investigate also the impact of past Antarctic ice sheet changes on local summer temperatures and consequently on TAC records.*

Figure 1B: the magnitude of EDC glacial-interglacial temperature change is uncertain, as per the discussion in the paper later on. It may be better to just plot d2H or d18O instead.
Done. We show now δD in the revised Figure 1.

Line 114 -115: I understand what is meant here, but the language is imprecise as there are not different "types" on insolation. Maybe rephrase to: "Comparing a proxy record with insolation is not necessarily straightforward because different metrics of insolation exist and their relationship with climate is not always clear"
Thanks for the proposition. It has been changed to "metrics".

Line 126: perhaps remind the reader that 75 degrees S is the latitude of EDC.
Done, the sentence has now been changed.

*Line 166: To avoid these assumptions, we propose to use a simpler and independent insolation index in the present work: the mean insolation over the astronomical half-year summer at 75°S (the latitude of EDC).*

Figure 2: Please add a third row of panels in which you compare the half-year summer insolation to the modeled half-year summer temperature. That way the reader can evaluate the relationship between these two key parameters better
The two curves can now be compared in Figure 3 from the revised manuscript (panel (e)).

Line 172-175: is it possible to evaluate how well LOVECLIM simulates the cited reconstructed seasonal temperatures from West Antarctica to validate the model simulations?
In the Figure R1 below we plot our simulated summer and winter temperatures at the WDC site against Fig.2 of Jones et al 2022. Our simulated summer temperature compares well with the reconstructed summer temperature in both the trend and the magnitude of temperature change. Both show that summer temperature in west Antarctica had an increasing trend from early to mid-Holocene and reached a maximum at ~4 ka BP followed by a decreasing trend. Our simulated winter temperature is quite different from the reconstructed one especially in the early Holocene during which our model simulates a cooling trend like in the ORBIT simulation of HadCM3, but the reconstruction shows a weak warming trend. According to the GLAC1D and ICE-6G simulations of HadCM3, using different ice sheet configurations could significantly influence the simulated winter temperature at WDC. Another possible reason for the mismatch between our simulated winter temperature and the reconstruction is that our model has a relatively low sensitivity in response to $CO_2$ change so the simulated warming due to a small $CO_2$ increase in the early Holocene is too weak in our model, letting the orbital forcing dominate. A brief introduction of this comparison has been added in our revised manuscript and we have also added a figure in the Supplementary material (Supplementary Figure S2) showing the comparison of the simulated and reconstructed summer temperature anomaly at the site of West Antarctic Divide ice core (WDC). The simulated result is from the OrbGHGIce transient simulation using LOVECLIM1.3, the reconstructed data is from Jones et al. (2023).

*Line 227: The comparison of the LOVECLIM1.3 simulated summer temperature in West Antarctica with the one reconstructed by Jones et al (2023) shows that they compare well in both the trend and the magnitude of temperature change over the Holocene, both showing that the summer temperature in west Antarctica had an increasing trend from early to mid-Holocene and reached a maximum at ~4 ka BP followed by a decreasing trend (supplementary Fig. S2). This validates the LOVECLIM1.3 simulations in reproducing past summer temperature changes in West Antarctica.*

[Figure]

Figure R1: Top and bottom panels: Simulated summer and winter temperature anomalies at WDC by LOVECLIM 1.3 OrbGHGIce transient simulation; Two middle panels: Fig. 2 of Jones et al. 2022.

[Figure]

*Supplementary Figure S2. Comparison of the simulated (a) and reconstructed (b) summer temperature anomaly at the site of West Antarctic Divide ice core (WDC). The simulated result is from the OrbGHGIce transient simulation using LOVECLIM1.3, the reconstructed data is from Jones et al. (2023).*

Line 181-182: Please provide more justification for this statement. Is this based solely on the increased correlation coefficient?

The linear coefficient between TAC (filtered) and simulated summer temperature indicates that about 58% of the TAC variability observed at EDC over the last 440 ka is explained by the half-year summer temperature. Nevertheless, this statement is based solely on this correlation coefficient, but there are more factors affecting TAC, as mentioned in the paper. Please see also more explanation in our reply to main comment 1.

Line 186: "This is information…." (information is singular)
Done.

Line 185-191: Can you clarify this discussion? Is TAC just a proxy for summer temperature, or is summer temperature actually driving TAC variations? The discussion here still assumes that TAC is driven by insolation directly.

We do not mean that TAC is driven by insolation directly. As explained in our reply to main comment 1, we mean that local summer insolation controls the temperature and the vertical temperature gradient in near-surface snow. Then, the surface snow structure is physically affected by changes in summer temperature. This surface snow structure change driven by summer temperature controls TAC. So, TAC can be used as a proxy for summer temperature. Furthermore, more details on the physical mechanisms are proposed in section 6. Our view is: insolation controls summer temperature, and summer temperature controls TAC, so TAC can be used as a proxy for summer temperature. This discussion has been clarified in the revised manuscript.

*Line 244: "Hence, we propose that the link between summer insolation and TAC variations exists through the summer temperature changes (see also section 6). Indeed, as proposed in previous studies (Lipenkov et al., 2011; Raynaud et al., 2007), the local summer insolation controls the near-surface snow temperature and the*

*vertical temperature gradients in snow. In turn, the latter affect the near-surface snow structure and consequently the porosity of the firn pores at close-off, i.e. the TAC."*

Figure 5: For comparison, can you please also plot the mean annual temperatures from the accelerated simulations that span the full 440ka period?
Done, we have now added a Figure S1 on the comparison of the annual mean temperature between the accelerated and non-accelerated simulations.

Line 217: "Response of Antarctic climate" (remove the 'a' at the end of Antarctica)
Done.

Line 221-222: In light of the discussion that follows, perhaps rephrase in more neutral language? For example: "One may also note that the magnitude of the temperature change between glacial and interglacial is significantly smaller in the model as compared to the reconstruction"
OK, we proposed to rephrase in the revised manuscript the sentence such as:

*Line 302: "One may also note that the magnitude of the temperature change between glacial and interglacial is significantly underestimated in the model as compared to the reconstruction".*

Line 223: It is my understanding that Antarctic elevation changes would exacerbate the problem, as EDC likely had lower elevation during the LGM.
It is an interesting point that deserves to be further studied. In principle, during glacial periods, the reduced surface accumulation rate leads to lower surface elevation (see Figure R2 below taken from the appendix of Raynaud et al., 2007). But there are also dynamical effects which make these reconstructions of surface elevation uncertain. Surface elevation changes affect our study in two different ways. First, TAC should be corrected for atmospheric pressure changes to get a record of porosity at close-off. Some of these atmospheric pressure changes are due to variations in surface elevation, another part might be due to change of atmospheric conditions (like the temperature of the air column). Because the reconstruction of surface elevation changes are uncertain, we have chosen to not correct for this effect. Second, surface elevation changes should also be taken into account in our simulation of summer or annual temperature at EDC by LOVECLIM. Unfortunately, in our climate model, the Antarctic temperature is fixed and we did not account for surface elevation changes. We could apply an a posteriori correction for surface elevation changes, but because the temperature variations (either annual of in summer) are probably under-estimated, applying this a posteriori correction would have a too strong influence. Finally, we should note that these two elevation corrections (for TAC and for the climate model) go in the same direction: during glacials, a corrected summer temperature would be warmer, and the corrected TAC of the ice would be smaller, so there is a chance that these two corrections would cancel each other and that the overall correlation between TAC and modeled summer temperature would not be much affected.
We have now included this discussion in the revised manuscript:

*Line 253: In our study, we did not account for variations in surface elevation. In principle, during glacial periods, the reduced surface accumulation rate leads to lower surface elevation (Raynaud et al., 2007). But there are also dynamical effects which make these reconstructions of surface elevation uncertain. Surface elevation changes affect our study in two different ways. First, TAC should be corrected for atmospheric pressure changes to get a record of porosity at close-off. Some of these atmospheric pressure changes are due to variations in surface elevation, another part might be due to change of atmospheric conditions (like the temperature of the air column). Because the reconstruction of surface elevation changes are uncertain, we have chosen to not correct for this effect. Second, surface elevation changes should also be taken into account in our simulation of summer or annual temperature at EDC by LOVECLIM. Unfortunately, in our climate model, the Antarctic temperature is fixed and we did not account for surface elevation changes. We could apply an a posteriori correction for surface elevation changes, but because the temperature variations (either annual or in summer) are probably under-estimated, applying this a posteriori correction would have a too*

*strong influence. Finally, we should note that these two elevation corrections (for TAC and for the climate model) go in the same direction: during glacials, a corrected summer temperature would be warmer, and the corrected TAC of the ice would be smaller, so there is a chance that these two corrections would cancel each other and that the overall correlation between TAC and modeled summer temperature would not be much affected.*

[Figure]

Figure R2. Simulated elevation changes at EDC from Raynaud et al. 2007.

Line 239-251: This is really excellent, and very instructive. What is the role of winter temperature? Is it fully controlled by GHG, such that in the annual average both GHG (winter) and ORB (summer) show up?

Thanks for this inspiring question. The wavelet analysis of the simulated mean half-year winter temperature at EDC shows very strong ~20-kyr cycle and an obvious ~100 kyr cycle, but the ~40-kyr cycle is very weak. It is illustrated in a new figure that is added to the revised manuscript (Revised Figure 8). The 100-kyr cycle results most probably from the effect of GHG. The very strong 20-kyr cycle but very weak 40-kyr cycle is quite intriguing. As far as insolation is concerned, the low-latitude insolation is dominated by the ~20-kyr precession cycle. As the solar energy received in Antarctica is very weak during local winter, the strong 20-kyr cycle in the simulated winter temperature could reflect a strong effect of the low latitude climate on the Antarctica temperature during austral winter, via for example meridional ocean and atmosphere heat transport. Revised Figure 8b shows a high, negative correlation between the simulated EDC winter temperature and precession. This indicates that the EDC winter temperature is strongly affected by boreal summer insolation in low latitudes (small precession parameter leads to high boreal summer insolation and vice versa). It is also shown in Yin and Berger (2012) that during some interglacials such as Marine Isotope Stages 5e, 15 and 17 which are characterized by very strong boreal summer insolation, a strong warming could be induced over Antarctica during austral winter, a warming which is even stronger than in many other regions due to polar amplification.

[Figure]

*Revised Figure 8. (a) Continuous wavelet transform of the simulated mean half-year winter temperature at the EDC site from the 10x accelerated OrbGHG simulation, and (b) Correlation between this winter temperature and precession. Low-pass filtered >12 kyr is applied on the winter temperature raw data.*

In our manuscript, the relative effects of insolation, GHG and NH ice sheets on the summer and annual temperature at EDC have been studied using the Orb, OrbGHG and OrbGHGIce experiments for the period 133-75 ka. The same experiments are used here to analyze their effects on the winter temperature. Figure R10 below shows that similar to what happens to the summer temperature, orbital forcing plays a dominant role also on the winter temperature at EDC. As explained above, the winter temperature at EDC is actually strongly driven by precession and boreal summer insolation, so on precession timescale, the orbitally-induced temperature variation in winter is in anti-phase with the summer temperature which is strongly driven by austral summer insolation. This anti-phase relationship leads to a strong weakening of the orbital signal especially the precession signal in the mean annual temperature (see Figure R2 below), making the effect of GHG and ice sheets more pronounced and leading to strong glacial cycles in the mean annual temperature.

[Figure]

Figure R3: (a) Mean half-year summer (October to March) temperature, (b) mean half-year winter (April-September) temperature and (c) mean annual temperature at the EDC site from the LOVECLIM1.3 transient simulations without acceleration for the period 133-75 ka. Black curves are the results from the Orb simulation under only orbital forcing; blue curves are the difference between OrbGHGIce and Orb simulations, showing the joint effect of GHG and ice sheets.

The discussions are now also added in the revised manuscript and simulated winter temperature changes are now shown in revised Figure 7:

*Line 336: To better understand the difference between the summer and annual mean temperature, the simulated winter temperature is also analyzed. The wavelet analysis of the half-year winter temperature shows very strong ~20-kyr cycle and an obvious ~100-kyr cycle, but the ~40-kyr cycle is very weak (Fig. 9a). The 100-kyr cycle results most probably from the effect of GHG. The very strong 20-kyr cycle but very weak 40-kyr cycle is quite intriguing. As far as insolation is concerned, the low-latitude insolation is dominated by the ~20-kyr precession cycle. As the solar energy received in Antarctica is very weak during local winter, the strong 20-kyr cycle in the simulated winter temperature could reflect a strong effect of the low latitude climate*

*on the Antarctica temperature during austral winter, possibly via meridional oceanic and atmospheric heat transport. Fig. 8b shows a high, negative correlation between the simulated EDC winter temperature and precession. This indicates that the EDC winter temperature is strongly affected by boreal summer insolation in low latitudes (small precession parameter leads to high boreal summer insolation and vice versa). It is also shown in Yin and Berger (2012) that during some interglacials such as Marine Isotope Stages 5e, 15 and 17 which are characterized by strong boreal summer insolation, a strong warming could be induced over Antarctica during austral winter, a warming which is even stronger than in many other regions due to polar amplification. Similar to what happens to the summer temperature, orbital forcing plays a dominant role also on the winter temperature at EDC (Fig. 7c). As explained above, the winter temperature at EDC is actually strongly driven by precession and boreal summer insolation, so on precession timescale, the orbitally-induced temperature variation in winter is in anti-phase with the summer temperature which is strongly driven by austral summer insolation. This anti-phase relationship leads to a strong weakening of the orbital signal especially the precession signal in the mean annual temperature (Fig. 7d), making the effect of GHG and ice sheets more pronounced and thus leading to strong glacial cycles in the mean annual temperature.*

[Figure]

*Revised Figure 7. Effect of insolation, GHG and NH ice sheets on the summer, winter and annual temperature at EDC. (a) CO2 concentration (blue, Lüthi et al., 2008) and NH ice volume anomaly as compared to pre-industry (green, Ganopolski and Calov, 2011), (b) simulated mean half-year summer (October to March) temperature, (c) simulated mean half-year winter (April-September) temperature, and (d) simulated annual mean temperature from the Orb, OrbGHG and OrbGHGICE experiments. The results of the LOVECLIM1.3 transient simulation without acceleration for the period 133-75 ka are used.*

Line 249: The GHG effect is 1 degree on both mean-annual and summer temperatures, so it is actually the same. Perhaps better to phrase it as the *relative* importance of GHG is bigger on annual-mean.
Thanks for the suggestion. It is indeed better to use "relative". The sentence has been changed to

*Line 328: "These results clearly show that as compared to insolation, GHG and NH ice sheets have relatively weaker effect on the summer temperature but they have relatively stronger effect on annual mean temperature".*

Fig, 6: The model appears to simulate a seasonal temperature cycle of around 16 degrees (twice the offset between summer and annual-mean). The observed seasonal cycle is closer to 40 degrees (from -25 in summer to -65 in winter). Please comment. How well does the model capture the seasonal variations in the modern day?

Please see our reply to main comment 4. The model captures quite well the seasonal temperature variations at EDC. New text and figures have been added in section 2.2 to comment on the performance of the model for the modern seasonal temperature cycle at EDC.

Section 6: can you me more clear in your usage of the words snow and firn? How do you define these? Except for the very fresh surface snow, anything older than 1 year I would call firn. But it appears you use the terms differently.
Indeed, there are no generally accepted definitions of snow and firn in the literature, and often these two terms are used interchangeably. Some papers on snow/firn metamorphism use only the term "firn" (e.g. Alley, 1987) while others, conversely, use only "snow" (e.g. Maeno & Ebinuma, 1983). The definition adhered to by the reviewer is more commonly found in the American literature (see, for example, Cuffey & Paterson, 2010, The Physics of Glaciers). However, in our work we follow the definition introduced by Anderson and Benson (1963), who associate snow and firn (névé in the original) with two different stages in the transformation of dry snow into bubbly ice. These two stages differ in the mechanisms that dominate the densification process: the particle rearrangement controlled by linear-viscous grain-boundary sliding (GBS) dominates in snow, whereas the power-law creep (PLS) dominates in further densification of firn. The snow-firn transition, viewed as a transition between the two densification regimes, is identified by the bending of the density-depth profile, which indicates a decrease in compaction rate and is usually observed when the critical snow density, equal to about 550 kg/m3 (which corresponds to a relative density of $D_0 = 0.6$ and is not a constant!), is reached. At this density, the number of contacts per grain approaches 6–7, thus making the sliding impossible. Recognition of the critical density introduces the possibility of making a physical distinction between snow and firn, or névé (Anderson & Benson, 1963). This conceptual framework has been adopted in a number of papers dealing with the physical modeling of snow-to-ice transformation (e.g. Arnaud et al, 1998, 2000) and the results of these works have been used in Raynaud et al. (2007) and Lipenkov et al. (2011) as a basis for discussing a possible mechanism of the effect of summer temperatures on the pore volume of firn at the close-off depth. We have clarified this in the introduction of the revised manuscript as such:

*Line 50: An imprint of local insolation changes has been evidenced in tracers which are measured in the air trapped in polar ice core. Indeed, air bubbles close-off from the surrounding atmosphere and they become trapped in ice, an air tight material resulting from the densification and diagenis of the snow deposited at the surface. These processes of densification and diagenis take place in the upper layer at the surface of the ice sheet (between 60-120 m-deep typically), which is characterized by an open porosity to the atmosphere and by two successive stages, snow and then firn, associated with different densification regimes (Anderson and Benson, 1963).*

Line 273: "ice grains" instead of "snow grains"?
Done.

Line 274: "upper few meters of THE snow column"
Done.

Line 278: the critical density you refer to here is around 550 kg/m3, correct? Or do you mean the critical density for pore closure? Please specify for clarity. What depth is this at EDC?
Yes, the first is correct. We also felt that the whole sentence was a bit misleading. To make the text clearer, we have changed the text in the revised manuscript to:

*Line 391: According to the model proposed by Arnaud (1997) for the Antarctic ice sheet, the pore volume at close-off, Vc, should increase with the mean annual surface temperature through the competing densification mechanisms: higher temperature leads to an increase in the relative critical density at the transition between snow and firn (at EDC the critical density is reached at a depth of about 25 m below surface), which in turn implies a greater proportion of the ice-grain edges occupied by pores at close-off, and hence a larger Vc.*

Line 279-280: is it no correlation or a poor correlation? Has to be either, can't be both.
It is a poor correlation. It has been changed

Line 281: there is no surface temperature record. Do you mean a correlation between TAC and the modeled summer temperature
Yes, we mean a correlation between TAC and the simulated mean summer surface temperature. It has been modified.

*Line 397: In contrast we observe a strong anti-correlation between TAC and the simulated mean surface summer temperature.*

Line 283: what is the number of pores per grain? Simply the ratio of the nr. of pores to the number of ice crystals?
Yes. We modified the text in the revised version.

*Line 399: Indeed, TAC increases with the ratio of the number of pores to the number of ice crystals at close-off.*

Line 284: do you mean the critical density of firn?
The critical density at the transition between snow and firn. Please see our replies above.

Line 284: This transition between GBS and PLC occurs at around 550 kg/m3, correct? Please specify for clarity. What depth is this at EDC?
Yes, this is correct. Please see our explanations and replies above.

Line 304-307: See my main comment, I would appreciate more clarity on what the authors think the causal relationships between insolation, summer temperature, and TAC are.
The causal relationships between insolation, summer temperature and TAC could be explained by that (1) local summer insolation is controlling the near surface summer temperature conditions, (2) this summer temperature conditions affect the near surface snow structure and then (3) the continuous nature of snow transformation until the trapping of the air in ice, i.e. TAC. So, the mechanism for the EDC TAC-summer temperature relationship is that insolation controls summer temperature and summer temperature controls TAC.
We have now rephrased the first paragraph of the conclusion as followed:

*Line 412: The lack of seasonal temperature reconstruction on Antarctica hampers a good understanding of the forcing and mechanism of climate changes over this climatically sensitive region. In this study, we revisit the TAC record measured in the EDC ice core covering the last 440 ka. We show that it is dominated by a 40-kyr periodicity and is anti-correlated with the local mean insolation over the astronomical half-year summer. In order to investigate further this link between local summer insolation changes and TAC variations, we look*

*into the correlation between the EDC TAC record and simulated local summer temperature changes by the LOVECLIM1.3 model. We evidence also an anti-correlation between those two independent variables. We explain the anti-correlations between local summer insolation/temperature and the EDC TAC by proposing that (1) the local summer insolation is controlling the development of strong temperature gradients in the near surface snow during the summer, (2) those summer temperature gradients are then modifying the surface snow structure and eventually (3) these snow structure changes propagate through the firn during the densification process down to the close-off depth where they impact the pore volume, i.e. the TAC of air bubbles (Lipenkov et al., 2011; Raynaud et al., 2007). These results points towards the fact that the EDC TAC record could be used as a unique proxy for local summer temperature. Future studies should investigate this relationship between TAC variations and local summer temperature changes in other ice core records drilled in Antarctica and Greenland.*

Line 308: Ultimately this insight does not come from the TAC record, but from the climate model. Do you agree? Without the climate model, one would not have interpreted TAC as a summer temperature proxy.
We do not totally agree. In Raynaud et al. (2007), TAC was already proposed to reflect summer temperature, and a reasonable physical mechanism could be proposed (also in the present manuscript) to explain how summer temperature could affect TAC. In this manuscript, the good comparison between TAC and the simulated summer temperature confirms the TAC could be a proxy for summer temperature. TAC itself has very different spectral characteristics from the δD which is used as an annual temperature proxy. So the statement of line 308 could be made based on both the TAC record and the model results.

**References:**

Alley, R., 1987. Firn densification by grain-boundary sliding: A first model. J. Phys. Colloques 48, DOI: 10.1051/jphyscol:1987135

Anderson, D.L. and G.S. Benson. 1963. The densification and diagenesis of snow: properties, processes and applications. In Kingery, VV. D., ed. Ice and snow: properties, processes, and applications. Cambridge, MA, M.LT.Press, 391- 41l.

Arnaud, L., V. Lipenkov, J.M. Barnola, M. Gay and P. Duval. 1998. Modelling of the densification of polar firn: characterization of the snow-firn transition. Ann. Glaciol., 26, 39-44.

Arnaud, L., J.-M. Barnola and P. Duval. 2000. Physical modeling of the densification of snow/firn and ice in the upper part of polar ice sheets. In Hondoh, T, ed., Physics of Ice Core Records, Hokkaido University Press, Sapporo, 285-305.

Lipenkov, V.Y., Raynaud, D., Loutre, M.F., Duval, P., 2011. On the potential of coupling air content and O2/N2 from trapped air for establishing an ice core chronology tuned on local insolation. Quaternary Science Reviews 30.

Maeno, N. and T. Ebinuma. 1983. Pressure sintering of ice and its implication to the densification of snow at polar glaciers and ice sheets. J. Phys. Chem., 87(21), 4103-4110.

Raynaud, D., Lipenkov, V., Lemieux-Dudon, B., Duval, P., Loutre, M.-F., Lhomme, N., 2007. The local insolation signature of air content in Antarctic ice. A new step toward an absolute dating of ice records. Earth Planetary Science Letters 261, 337-349, doi:310.1016/j.epsl.2007.1006.1025.

Yin, Q.Z. and Berger, A., 2012. Individual contribution of insolation and $CO_2$ to the interglacial climates of the past 800 000 years. Climate Dynamics, 38, 709-724.

**Reply to Review 2**

We thank Reviewer 2 for the constructive and insightful comments. Here is our point-by-point reply. The review comments are in black and our replies are in green. ==Changes are indicated by yellow highlights in the revised manuscript version== that indicates the modifications made according to the reviewer's comment.

**General comments:**
The manuscript explores the idea that there are no good reconstructions of summer temperature and suggests that TAC could be used as a proxy for summer temperature in Antarctica. The authors use a previously measured record of TAC from the EDC ice core, which spans the last 440 ka. They use the EDC TAC data to compare with a climate model simulation of summer temperature to make the case that TAC could be used as a summer temperature proxy.
Overall, the manuscript is good, and the idea for using TAC as a summer temperature proxy is important, especially if it could be shown to also be used on other ice cores (a subject for future research). I recommend its publications after the authors consider the below comments:

**Specific comments:**
Overall, it is difficult to follow if the authors main point is that the TAC is controlled by summer insolation or summer temperature, or both. The authors clearly describe that both summer insolation and summer temperature are well anti-correlated with TAC. To make this clearer, I would recommend reorganizing the conclusion, highlighting their main argument at the beginning of the first paragraph.
Reviewer 1 made the same remark. Here we use the same reply to Reviewer 1's main comment 1:
*We mean option (a). The link between insolation and TAC is through summer temperature. In the introduction (p.2, lines 59-62) of our manuscript we have explained that the anti-correlation between local summer insolation and TAC can be attributed to a mechanism where the local summer insolation is controlling the near-surface snow temperature and temperature gradients during summer time, which affects the near surface snow structure and hence TAC. In section 6 we discuss the possible mechanisms linking TAC and local summer temperature. We propose a mechanism based on snow/firn physics, which could explain the strong anti-correlation observed between TAC and the mean summer surface temperature. Nevertheless, a numerical model, which takes into- account the successive mechanisms involved between the surface snow and the closure of pores, is still required. Such model would explain that time periods with higher summer insolation and summer temperature will promote a coarser-grained snow structure, a lower critical density of snow and then a reduced TAC at pore closure.*

We have now stress more about this point in the revised manuscript in several places and reorganized the conclusion accordingly.

*In the abstract:*
*Line 21: We propose that (1) the local summer insolation controls the local summer temperature, (2) the changes in the snow structure are affected by the summer temperature gradient at the near surface and (3) those snow structure changes are controlling eventually the pore volume at the bubble close-off and hence, the TAC.*

*In section 4:*
*Line 244: Hence, we propose that the link between summer insolation and TAC variations exists through the summer temperature changes (see also section 6). Indeed as proposed in previous studies (Lipenkov et al., 2011; Raynaud et al., 2007), the local summer insolation controls the near-surface snow temperature and the vertical temperature gradients in snow. In turn, the latter could affect the near-surface snow structure and consequently the porosity of the firn pores at close-off, i.e. the TAC.*

*In the conclusion:*

*Line 412: The lack of seasonal temperature reconstruction on Antarctica hampers a good understanding of the forcing and mechanism of climate changes over this climatically sensitive region. In this study, we revisit the TAC record measured in the EDC ice core covering the last 440 ka. We show that it is dominated by a 40-kyr periodicity and is anti-correlated with the local mean insolation over the astronomical half-year summer. In order to investigate further this link between local summer insolation changes and TAC variations, we look into the correlation between the EDC TAC record and simulated local summer temperature changes by the LOVECLIM1.3 model. We evidence also an anti-correlation between those two independent variables. We explain the anti-correlations between local summer insolation/temperature and the EDC TAC by proposing that (1) the local summer insolation is controlling the development of strong temperature gradients in the near surface snow during the summer, (2) those summer temperature gradients are then modifying the surface snow structure and eventually (3) these snow structure changes propagate through the firn during the densification process down to the close-off depth where they impact the pore volume, , i.e. the TAC of air bubbles (Lipenkov et al., 2011; Raynaud et al., 2007). These results points towards the fact that the EDC TAC record could be used as a unique proxy for local summer temperature. Future studies should investigate this relationship between TAC variations and local summer temperature changes in other ice core records drilled in Antarctica and Greenland.*

Line 190 the authors state that TAC can be used as a proxy for summer temperature, based on the strong anti-correlation with modeled summer temperature. Then in section 6 the authors highlight that summer temperature influencing firn metamorphism is only a low-accumulation site phenomenon. Does this mean that TAC could not be used as a temperature proxy in Greenland? If this is the case, maybe the language about using TAC as a proxy for summer temperature is too strong for their results. While their result is interesting, before saying that TAC is a proxy for summer temperature, the relationships should be verified at multiple ice core sites.

Thank you for this important question. We have now made it clear that our conclusion applies to the EPICA Dome C but that it has to be demonstrated for other sites in Antarctica and in Greenland. Note that this is an on-going work led by a PhD student at IGE. Going in this direction and to be more precise we have changed the title of our paper into:

*"Past local summer temperature changes revealed by the Total Air Content record from the Antarctic EPICA Dome C ice core".*

We also made the following changes:

*Line 248: "The good correlation between the two independent climate variables, TAC measured from ice cores and summer temperature simulated by the LOVECLIM1.3 model, indicates that the EDC TAC record can be used as a proxy for local summer temperature. The relationship between the TAC record from the EDC ice core and local summer temperature should be further investigated in other ice cores from Antarctica and Greenland".*

*Line 425: "Future studies will investigate this relationship between TAC variations and local summer temperature changes in other ice core records drilled in Antarctica and Greenland."*

Line 49: What is the difference between V (air content) and TAC (total air content)? A differentiation of what the authors mean by V vs TAC is required.

We wrote (lines 49-50*):" During previous works V (for air content) and TAC (for Total Air Content) have been indifferently used for designating the same property. In this work we are using TAC, which is usually used in the recent works."*

So, there is no difference between the use of V and TAC, both designating the same property. As said, in this work we are using TAC, which is currently used in the recent works. We have modified a sentence in the revised manuscript as followed:

*Line 66: During previous works V (for air content) and TAC (for Total Air Content) have been interchangeably used for designating the same property. In this work we are using TAC, following other recent work.*

Line 84: What is a 10x acceleration, and why is it important in this context? I'm guessing the 10x model was used to save resources, but I recommend that more information is presented about why the accelerated models were used instead of the non-accelerated simulations.

Indeed the 10x acceleration was used to save computing resources and time. This technique can be explained by the text below cited from page 4 of Yin and Berger (2015) which is referred in our manuscript: "*Although being a model of intermediate complexity, LOVECLIM remains still costly for transient experiments, particularly when 5 interglacials and 10 transient simulations are considered. An acceleration technique similar to Lorenz and Lohmann (2004) was therefore used to speed up the simulations and reduce the computational costs. An acceleration factor of 10 is used, which means that at the end of each year of the simulation, the astronomical parameters and GHG concentration are advanced by 10 years. In such a case, the actual length of the simulation is reduced by 10 times. For example, a 20,000-year long simulation only needs 2000 model years. To test the impact of such an acceleration technique on our transient simulations, a non-accelerated experiment and a 10-time accelerated one have been done for two interglacials MIS-5 and MIS-13. Our results showed that the acceleration method has little impact on the surface air temperature and precipitation. However, the response of the deep-ocean temperature is delayed by 2-3 ka in the accelerated simulations as compared to the non-accelerated ones, similar results being observed also in other studies (Timm and Timmermann, 2007; Ganopolski and Calov, 2011). A detailed analysis made by Timm and Timmermann (2007) shows that a 10-time acceleration leads to a delayed response of the temperature only in the inner ocean. As here we are mainly interested in surface climates, the 10-time acceleration technique would not alter our conclusion about the phasing between the surface temperatures of different regions.*"

In our manuscript, the results of the 10x accelerated simulation are used for the entire 440 ka, but non-accelerated simulations for five glacial-interglacial episodes are also used to compare with the accelerated one. The comparison in Supplementary Figure S1 (see below) shows that the 10x acceleration technique does not affect the summer and annual temperature that were discussed in our manuscript. In our revised manuscript, we are providing more information on these aspects:

*Line 109: Although LOVECLIM1.3 is classified as an EMIC model, its complexity is high for this kind of models and its ocean component is a full general circulation model, so it remains challenging to run full transient simulations with this model. We therefore first performed a transient simulation with 10x acceleration covering the last 800 ka, which allows to compare the simulated local summer temperature with the TAC record over the entire last 440 ka. In this simulation the variations of orbital forcing and GHG were considered, and the global ice sheets were fixed to their pre-industrial condition. Using the same model and the same acceleration technique, it has been shown that 10x acceleration has a significant impact on deep ocean temperature, but it has no major impact on surface temperature (Yin and Berger, 2015). This is further confirmed in our study where the Antarctic summer and annual mean temperature changes of the 10x acceleration simulations are matching well with that of the non-accelerated simulations (Supplementary Fig. S1).*

[Figure]

*Supplementary Figure S1. Comparison of 10x accelerated (purple) and non-accelerated (blue) simulations for the simulated mean half-year summer (October to March) temperature (a) and mean annual temperature (b) at EDC. Both are OrbGHG simulations without considering ice sheet changes. 1000-year running mean is plotted.*

Line 111: The sentence beginning "This spectral characteristic…" suggests that orbital astronomical forcing drives changes in TAC. The studies correlations do show a link between TAC and orbital patterns, but I think this sentence is misleading, and should be changed to reflect that TAC is only correlated with astronomical forcing, not actually caused by it. Later, I believe the authors make the argument that temperature gradients are what is causing TAC to vary, not insolation alone.
We have now rephrased the sentence in the revised manuscript such as:

*Line 151: Overall, this spectral characteristic illustrates that the variations of TAC are strongly correlated by the astronomical forcing and could be linked to insolation changes (Lipenkov et al., 2011; Raynaud et al. 2007).*

Line 127: I like the description of the astronomical half-year, and it is intuitive as to why it would be used as opposed to ISI.
Thank you.

Line 151: Here the authors say that TAC can be considered a proxy driven by mean local summer insolation based on the correlation. I'm a bit unclear here, because the correlation is higher (0.58) between TAC and summer temperature than between TAC and insolation (0.39). Do the authors mean that insolation drives summer temperature drives TAC? Or are they referring to previous works results using integrated summer insolation? If so, what was the correlation coefficient in that instance?
This sentence was misleading, we decided to remove it in the revised manuscript.

However, to clarify what we mean overall, we propose that TAC is driven by summer temperature which is in turn mainly driven by local summer insolation. However, as explained in our reply to Reviewer 1's main comment 2, although local summer insolation is a main factor controlling summer temperature, other factors such as GHG could also contribute. This is why the correlation between TAC and summer temperature is higher than the correlation between summer temperature and insolation. We have made the following changes in the revised manuscript in order to clarify our view:

*In the abstract:*
*Line 24: We propose that (1) the local summer insolation controls the local summer temperature, (2) the changes in the snow structure are affected by the summer temperature gradient at the near surface and (3) those snow structure changes are controlling eventually the pore volume at the bubble close-off and hence, the TAC.*

*In section 4:*
*Line 244: Hence, we propose that the link between summer insolation and TAC variations exists through the summer temperature changes (see also section 6). Indeed as proposed in previous studies (Lipenkov et al., 2011; Raynaud et al., 2007), the local summer insolation controls the near-surface snow temperature and the vertical temperature gradients in snow. In turn, the latter could affect the near-surface snow structure and consequently the porosity of the firn pores at close-off, i.e. the TAC.*

*In the conclusions:*
*Line 412: The lack of seasonal temperature reconstruction on Antarctica hampers a good understanding of the forcing and mechanism of climate changes over this climatically sensitive region. In this study, we revisit the TAC record measured in the EDC ice core covering the last 440 ka. We show that it is dominated by a 40-kyr periodicity and is anti-correlated with the local mean insolation over the astronomical half-year summer. In order to investigate further this link between local summer insolation changes and TAC variations, we look into the correlation between the EDC TAC record and simulated local summer temperature changes by the LOVECLIM1.3 model. We evidence also an anti-correlation between those two independent variables. We explain the anti-correlations between local summer insolation/temperature and the EDC TAC by proposing that (1) the local summer insolation is controlling the development of strong temperature gradients in the near surface snow during the summer, (2) those summer temperature gradients are then modifying the surface snow structure and eventually (3) these snow structure changes propagate through the firn during the densification process down to the close-off depth where they impact the pore volume, , i.e. the TAC of air bubbles (Lipenkov et al., 2011; Raynaud et al., 2007). These results points towards the fact that the EDC TAC record could be used as a unique proxy for local summer temperature. Future studies should investigate this relationship between TAC variations and local summer temperature changes in other ice core records drilled in Antarctica and Greenland.*

Line 149, 150, 170: The authors use the term 'correlated' when I think they mean anti-correlated, or negatively correlated. This needs to be clear, as it can cause confusion. I recommend all instances are reviewed.
Yes, sorry for the confusion, we meant anti-correlated. This is corrected now in all appropriate places.

Line 177: Figure 2 does not show any correlation between summer temp and summer insolation. Recommend adding this to figure 2 or deleting this line.
This is now shown in Revised Figure 3, panel (e).

[Figure]

*Revised Figure 3. Comparison of TAC record with (a) mean insolation during astronomical half-year summer at 75°S calculated using the solution of Berger and Loutre (1991) and with (c) simulated mean half-year summer (October to March) temperature at EDC. Their corresponding linear regression analyses are shown in (b) and (d). (e) Comparison of half-year summer insolation with simulated mean half-year summer temperature and corresponding linear regression in (f). Low-pass filtered >12 kyr is applied on the TAC, insolation and summer temperature raw data before comparison. Note that the y axis for TAC is reversed on the left panels to ease the visual comparison.*

Line 280: What does "no (poor) correlation" mean? Please give a correlation coefficient. You reference section 5, but maybe figure 1 would be a better reference?
We mean poor correlation. The text has been changed accordingly. You are right that Figure 1 is a better reference. It has been modified.

Line 282: What does it mean to 'affect negatively" the critical density? Does this mean the other factor decreases the critical density?
Yes, this sentence needs rewording. The modified text reads:

*Line 398: "This observation suggests that summer temperature has an inverse effect on Vc compared to the mean annual temperature".*

Line 308: I thought the comparison came from dD and modelled summer temperature, not TAC and dD?

We do mean the comparison between TAC and δD. A simple comparison of their spectral characteristics already indicates that their major drivers are different.

**Technical Comments:**
 Line 3 – insert ",a" before "..tracer"
Done.

Multiple places EDC is referred to as Dome C. Recommend using EDC throughout the paper.
Done.

Line 155 – Recommend starting new paragraph here, where you start to discuss orbital tuning.
Done.

Figure 3 units – Are your units on the left hand side correct? I think they should be the same order of magnitude as the right side.
Yes they are. We calculate ISI 380 as a flux (in W.m-2) following the same definition as the one provided in Eicher et al. (2016). We added a note in the revised manuscript.

*Line 201: Caption of Figure 4. (a) Comparison of ISI 380 (blue) with the mean insolation during astronomical half-year summer (red), (b) evolution of the phase delay (purple) between the two insolation curves filtered in the 15-46 kyr band. ISI 380 is represented here as a flux following the definition given in Eicher et al. (2016).*

Line 211 – Strange spacing issue here.
Corrected.

Line 290: Recommend deleting the second 'here' in the sentence.
Done.

Line 312: Missing a word. Perhaps "Our transient simulation which allows us…"
Done.

**Additional references:**

Yin, Q.Z. and Berger, A., 2015. Interglacial analogues of the Holocene and its natural near future. Quaternary Science Reviews 120, 28-46.

---

## Editor Decision (ED1)

[revised manuscript text omitted]